# Essential Protective Role of Catalytically Active Antibodies (Abzymes) with Redox Antioxidant Functions in Animals and Humans

**DOI:** 10.3390/ijms23073898

**Published:** 2022-03-31

**Authors:** Anna S. Tolmacheva, Georgy A. Nevinsky

**Affiliations:** Institute of Chemical Biology and Fundamental Medicine, SB of the Russian Academy of Sciences, 630090 Novosibirsk, Russia; anny_@mail.ru

**Keywords:** autoimmune diseases, SLE, multiple sclerosis and other pathologies, hematopoietic stem cells differentiation, non-enzymatic and enzyme antioxidants, catalytic antibodies with antioxidants with redox activities

## Abstract

During the life of aerobic organisms, the oxygen resulting from numerous reactions is converted into reactive oxygen species (ROS). Many ROS are dangerous due to their high reactivity; they are strong oxidants, and react with various cell components, leading to their damage. To protect against ROS overproduction, enzymatic and non-enzymatic systems are evolved in aerobic cells. Several known non-enzymatic antioxidants have a relatively low specific antioxidant activity. Superoxide dismutases, catalase, glutathione peroxidase, glutathione S-transferase, thioredoxin, and the peroxiredoxin families are the most important enzyme antioxidants. Artificial antibodies catalyzing redox reactions using different approaches have been created. During the past several decades, it has been shown that the blood and various biological fluids of humans and animals contain natural antibodies that catalyze different redox reactions, such as classical enzymes. This review, for the first time, summarizes data on existing non-enzymatic antioxidants, canonical enzymes, and artificial or natural antibodies (abzymes) with redox functions. Comparing abzymes with superoxide dismutase, catalase, peroxide-dependent peroxidase, and H_2_O_2_-independent oxidoreductase activities with the same activities as classical enzymes was carried out. The features of abzymes with the redox activities are described, including their exceptional diversity in the optimal pH values, dependency and independence on various metal ions, and the reaction rate constants for healthy donors and patients with different autoimmune diseases. The entire body of evidence indicates that abzymes with redox antioxidant activities existing in the blood for a long time compared to enzymes are an essential part of the protection system of humans and animals from oxidative stress.

## 1. Introduction

Antibodies (Abs) were described as specific proteins produced by the immune systems of different organisms, which possess the exclusive function of different antigens binding and providing an immune response. Abs can act similarly to classical enzymes in recognizing various molecules in this classical conception. It was believed that in contrast to canonical enzymes, however, immunoglobulins cannot catalyze any chemical reactions. For the vast majority of Abs, this idea is correct. For many researchers, including immunologists, it was very unexpected and unpredictable that Abs, in addition to the classical function of antigens binding, are possessed of catalytic activities similar to enzymes. Linus Pauling’s analysis (1946) led him to conclude that the enzyme’s active centers are maximally fitted not to substrates but to the chemical reactions transition states [1]. In addition, the sites of various antigens recognition by immunoglobulins are also maximally adjusted to their structure. However, it is impossible to obtain Abs against very short-lived transitional states of chemical reactions. Jencks (1969) proposed that antibodies with enzymatic activity can be obtained against chemically stable analogs of transition states of chemical reactions [2]. Thus, two scientists predicted the possibility of the existence of antibodies with enzymatic functions.

The general methods for generating monoclonal antibodies against stable analogs of transition states (SATS) with enzymatic activities were first described in 1985 [3]. This idea was implemented much later. Two scientific groups came up with how to obtain the first monoclonal Abs with catalytic activities using stable analogs of substrates containing instead of a carbon atom, a phosphorus atom: p-nitrophenylphosphorylcholine [4] and monoaryl phosphonate esters [5,6]. These catalytic Abs were named abzymes (ABZs, derived from antibody-enzyme) [7,8,9,10,11,12,13,14,15,16]. These catalytically active antibodies will be called artificial abzymes. These early results clearly showed that the complementarity of antibodies with stable analogs of transition states could be used as a general strategy for generating various catalytic Abs [4,5,6,7,8,9,10,11,12,13,14,15,16,17,18,19,20]. To date, the literature describes more than 200 different artificial ABZs against SATS with a variety of catalytic functions. At present, many different approaches to designing ABZs with different properties are known, including methods of site-directed mutagenesis and selective chemical modification [18,19]. A new type of abzyme can possess broader substrate specificity and catalyze many different chemical reactions at a rate comparable to classical enzymes with related activities or even higher. Moreover, artificial abzymes were created, for which there are no analogs among natural enzymes [18,19]. The transformation of the chorismate into the prefenate (Claisen rearrangement) is an essential step in the biological synthesis of aromatic amino acids in bacteria. Abzymes catalyzing the Claisen rearrangement were constructed. No enzymes were known until recently, when they were discovered catalyzing the Diels–Alder reaction. This reaction of a diene and an alkene leads to a cyclohexene product. Abzymes catalyzing the reaction of Diels–Alder were also obtained [18,19].

At present, artificial ABZs are mainly aimed at obtaining specific monoclonal abzymes as medicines. Cocaine hydrolyzing ABZs have been created [20,21]. Anti-cocaine catalytic Abs may be useful as an extra therapeutic cure for patients poisoned with this narcotic. Marihuana is one of the widely used prohibited narcotics in the world. ABZs catalyzed oxidation of 9-tetrahydrocannabinol, the main psychoactive component of marihuana, was reported [21]. There are several reviews devoted to the data on creating abzymes for medical purposes [22,23,24].

To understand the differences and similarities between classical enzymes and antibodies with redox functions, it is important to describe the features of oxidative stress and classical antioxidant enzymes.

## 2. Reactive Oxygen and Nitrogen Species (RONS) Formation and Oxidative Damage of Biomolecules

During the life of an aerobic organism, about 5% of oxygen consumed by tissues as a result of numerous reactions is converted into reactive forms of oxygen and nitrogen (RONS), such as singlet oxygen (^1^O_2_), superoxide anion radical (O_2_^•−^), hydroxyl radical (^•^OH), hydrogen peroxide (H_2_O_2_), nitroxyl radical (R_2_NO^•^), peroxynitrite (ONOO^−^), nitric oxide (NO), peroxide radical (ROO^•^), alkyl radicals (RC^•^HR), alkylperoxy radicals (RCOO^•^), and hypochlorite (ClO^−^), which are highly reactive [25,26,27,28]. 

The most significant reactive oxygen species (ROS) sources in eukaryotic cells are the mitochondrial respiratory chain, the cytochrome P450 microsomal system, and the metabolism of fatty acids in peroxisomes [29,30]. For the first time, the generation of superoxide radicals in mitochondria was reported more than four decades ago [31]. Usually, in the process of gradual four-electron reduction of molecular oxygen to water on the electron transport chain of mitochondria, a small number of electrons “leak”, leading to the formation of O_2_^•−^ or H_2_O_2_. In the case of structural and functional disorders of mitochondria, an increase in the non-physiological production of superoxide anion-radical oxygen and hydrogen peroxide is observed [32].

In addition to mitochondria, there are other cellular sources of the superoxide anion radical, for example, xanthine oxidase, a molybdenum-containing enzyme widely presented in various mammalian tissues. Xanthine oxidase realizes the oxidation of hypoxanthine to xanthine, and xanthine to uric acid, accompanied by the formation of a superoxide radical anion and hydrogen peroxide [33]. Despite their short lifetime, 10^−6^ s, both O_2_^•−^ and ^1^O_2_, have a large radius of action comparable to the cell size. Additional endogenous sources of O_2_^•−^ are neutrophils, eosinophils, and macrophages. Activated macrophages initiate the increase in oxygen uptake, resulting in various reactive oxygen species, including O_2_^•−^ and hydrogen peroxide [34].

Formed in the cell as a result of various processes, O_2_^•−^ turns to O_2_ and hydrogen peroxide (H_2_O_2_) by the enzyme superoxide dismutase (reaction 1) [35]. Hydrogen peroxide is an oxidizing agent and, being an electrostatically neutral molecule, can freely pass through membranes.
2O_2_^•−^ + 2H^+^ → O_2_ + H_2_O_2_(1)

Microsomes and peroxisomes containing a number of hydrogen peroxide-generating oxidases are also an important source of hydrogen peroxide in the cells. Thus, formed hydrogen peroxide is relatively unreactive, but in the presence O_2_^•−^ it can form extremely reactive hydroxyl radical ^•^OH through the Haber–Weiss reaction (reaction 2) [36].
O_2_^•−^ + H_2_O_2_ → O_2_ + ^•^OH + HO^−^(2)

In addition, ^•^OH radical is also formed in the case of interaction of hydrogen peroxide with transition metal ions, such as iron, copper, chromium, or cobalt via the Fenton reaction [37,38] (reaction 3).
Fe (II) + H_2_O_2_ → Fe (III) + ^•^OH + HO^−^(3)

Another source of reactive oxygen species in aqueous solutions is X-, γ-, or UV-irradiation [39]. During these processes, such forms of ROS as O_2_^•−^, H_2_O_2_, ^•^OH, HO^−^, and HO_2_^•^ are formed.

Myeloperoxidase, found in phagocytic cells, which using hydrogen peroxide catalyze the two-electron oxidation of chloride (Cl^−^) to hypochlorous acid (HOCl) (reaction 4).
H_2_O_2_ + Cl^−^ + H^+^ → HOCl + H_2_O(4)

This compound is a strong oxidizing agent that acts as a powerful antimicrobial compound. However, sustained high levels of HOCl may be one of the possible causes of some pathological conditions. For example, in the case of a number of chronic inflammatory diseases in which myeloperoxidase activity is increased, such as atherosclerosis, cancer, or pulmonary fibrosis, iron is oxidized by HOCl in the heme of proteins containing heme and is released in a free form [40], which, in turn, can lead to the formation of a hydroxyl radical (^•^OH) (reaction 5):ClO^−^ + Fe (II) + H^+^ → Fe (III) + Cl^−^ + ^•^OH(5)

Nitric oxide is formed in mammalian cells by enzymatic oxidation of the amino acid L-arginine to citrulline by NO synthases [41]. Nitric oxide molecules are quite long-lived and freely diffuse in biological media. Nitric oxide can react with superoxide radical anion to form highly toxic peroxynitrite ONOO^−^ [42] (reaction 6):O_2_^•−^ + NO → ONOO^−^(6)

In normally functioning cells, ROS are involved in processes necessary for vital body activity, such as the synthesis of biologically active substances, collagen metabolism, regulation of membrane permeability, formation of O_2_^•−^ in activated macrophages and neutrophils in the focus of inflammation [43], signaling transduction [44,45], and activation of enzymes necessary for the implementation of apoptosis. However, RONS can be dangerous because they are strong oxidants and react with various cell components, leading to their damage due to their high reactivity.

DNA is one of the targets for ROS. Free radicals, mainly ^•^OH, cause considerable damage to its structure, such as modifications of bases and sugar-phosphate backbone, chain breaks and DNA–DNA, DNA–protein cross-link formation, and exo- and endocyclic adducts’ appearance [46,47]. Such damages can lead to mutations and are also one of the causes of carcinogenesis and ageing [48].

Oxidation of proteins by RONS leads to many different types of covalent protein–protein cross-linked derivatives [28,37] and intra- or intermolecular cross-links due to the oxidation of sulfhydryl groups of cysteine residues (–S–S– cross-links) [49]. Peroxynitrite (ONOO^−^) is able to interact with sulfhydryl groups of proteins, nitrate amino acid residues of tyrosine and tryptophan, and oxidize methionine to form methionine sulfoxide. This modified tyrosine product has been found in Alzheimer’s and Parkinson’s diseases and patients with amyotrophic sclerosis [50]. 

Generated in aqueous solutions by the use of X-, γ-, or UV-rays ROS, such as ^•^OH, O_2_^•−^ and its protonated form, the hydroperoxyl radical (HO_2_^•^) can attack electrophilically both the main peptide chain and the side chains of peptides, polypeptides, and proteins, causing proteolytic cleavage [39,51].

Free radical oxidation of fatty acids leads to the formation of many highly toxic products for the cell. This process begins with the formation of a free carbon radical as a result of the interaction of polyunsaturated fatty acids of the side chains of membrane lipids with O_2_^•−^ or •OH radicals. The carbon radical, in turn, reacting with molecular oxygen, forms fatty acid peroxyl radicals, which, when interacting with the hydrogen of another nearest polyunsaturated fatty acid, forms lipid hydroperoxide and a new carbon radical. In the process of such a chain reaction, many products are formed: lipid hydroperoxides, aldehydes, ketones, epoxides, diene conjugates, and peroxide radicals. In the process of oxidative degradation of lipids, products such as malondialdehyde (MDA) or 4-hydroxy-2-nonenal (4-HNE), which are mutagens, are formed [52,53]. These compounds can react with nitrogenous bases, forming exocyclic adducts [54].

Interacting with the amino groups of lysine, the sulfhydryl groups of cysteine, and the imidazole group of the histidine of proteins, MDA and 4-HNE, can lead to the formation of intra- and interprotein cross-links, thereby causing structural damage to various proteins and enzymes [55]. Such damage leads to cellular dysfunction and ultimately to cell death. For example, HNE–protein adducts have been found in oncological, neurodegenerative, chronic inflammatory, and autoimmune diseases [55]. However, it is important to note that due to its ability to bind to functional proteins covalently, HNE is not only a toxic product of lipid peroxidation, but also a regulatory molecule involved in various biochemical processes [55,56].

If the RONS level in a cell reaches a critical value, a condition called oxidative stress takes place. This condition leads to destructive biochemical processes for the cell and is one of the causes of many diseases. Currently, overproduction of ROS is associated with oncological [57,58], mental [59,60], neurodegenerative [61], and cardiovascular diseases [62], type 2 diabetes mellitus [63], chronic obstructive pulmonary disease [64], autoimmune diseases [65,66], the aging process [67], osteoporosis [68], asthma [69], and gastrointestinal disorders [70].

## 3. Non-Enzymatic Antioxidants

Non-enzymatic antioxidants include compounds that have a relatively low specific antioxidant activity. Nevertheless, being present in high concentrations in the water and lipid phases, they could make an essential contribution to the overall antioxidant activity [71]. One example of such antioxidants is α-tocopherol, the most biologically active form of Vitamin E. It is a fat-soluble membrane-bound antioxidant that effectively interacts with peroxide radicals of essential fatty acids [72,73,74] or superoxide radical anion and prevents cell membranes from lipid peroxidation. This vitamin is also able to protect cells from the damage caused by the NO radical [75].

Vitamin C (l-ascorbate) is an important antioxidant involved in the inactivation of O_2_^•−^, ^•^OH, HO_2_^•^, and HOCl, as well as organic peroxides. Binding and inactivating free radicals, it protects low-density lipoproteins and other membrane lipids from oxidative damage [76]. It is mostly contained in bronchoalveolar secretion and cerebrospinal fluid [77]. In addition to neutralizing ROS, vitamin C is also involved in the reduction of other antioxidants, such as the α-tocopheryl radical. At the same time, in the presence of metal ions, for example, Fe (III), l-ascorbate can also act as a prooxidant since it transfers these metal ions to a reduced state and thus induces the formation of free radicals (Fenton reaction) [78]. Vitamin C is also involved in the regulation of apoptosis, gene expression, and other processes occurring in cells [79,80].

Vitamin A (retinol), such as vitamin E, is a fat-soluble compound with antioxidant activity. It prevents the conversion of sulfhydryl groups into disulfides and also effectively removes peroxyl radicals (ROO^•^), protecting membranes from lipid peroxidation [81]. At the same time, a number of by-products of a radical nature can be formed in this process, for example, the peroxyl radical of retinol. Thus, retinol can also act as a pro-oxidant [82,83]. Vitamin A affects the expression of genes responsible for the implementation of an effective antioxidant response [84], influences the processes of cell differentiation and proliferation, and may be involved in the regeneration of the α-tocopheryl radical. Thus, as antioxidants, vitamins A, C, and E act together and form the so-called antioxidant vitamin group. Their intake into the body reduces the risk of developing cancer, cataracts, and cardiovascular diseases [81].

Coenzyme Q_10_ reacts with oxygen radicals O_2_^•−^, ^•^OH, RO^•^, and ROO^•^, and is the main lipophilic antioxidant in mitochondria and platelets. Coenzyme Q_10_ in some cases also acts as a prooxidant: being oxidized and reduced in the process of electron transport along the mitochondrial respiratory chain, coenzyme Q_10_ and its radical can transfer electrons to O_2_ with O_2_^•−^ formation [85].

Another unique example of a non-enzymatic antioxidant is lipoic acid (LA). This compound is both water- and fat-soluble, due to which it is able to exhibit antioxidant activity both in the composition of cell membranes and in the cytosol. Lipoic acid enters the body from food and in the tissues is converted into a reduced form—di-hydrolipoic acid (DHLA). LA neutralizes ^•^OH radical, singlet oxygen, and hypochlorite. DHLA can neutralize O_2_^•−^ and ^•^OH radicals, hypochlorite, and peroxyl radicals [86]. In addition, LA and DHLA are able to chelate metals with variable valence, preventing the formation of ROS [87]. DHLA activates the activity of other antioxidants, such as α-tocopherol, vitamin C, glutathione, and coenzyme Q10, and modulates the activity of some transcription factors [88].

An essential component of maintaining redox homeostasis is SH-containing compounds—proteins and peptides, including cysteine and methionine. They play a significant role in cells from protecting hydroxyl radicals. In the blood of mammals, the main carriers of SH-groups are albumin forms, which account for up to 76% of the total mass of serum proteins [89]. Glutathione occupies a special place among SH-compounds [90,91]. Moreover, non-enzymatic antioxidants include chelators of variable valence ions, such as transferrins (serum transferrin, lactoferrin, ovotransferrin, melanotransferrin), ferritin, ceruloplasmin, metallothionein-like proteins, and uric acid). All these compounds can bind metal ions of variable valence and thereby prevent their participation in the decomposition reactions of peroxides with the formation of a reactive hydroxyl radical [92,93].

## 4. Enzymatic Antioxidants

To protect against ROS overproduction, enzymatic systems in aerobic cells are evolved. The most important members of the enzymatic system are superoxide dismutases (SODs), catalase, glutathione peroxidase (GPxx), glutathione S-transferase, the thioredoxin family (Texas) [94,95,96], and the peroxiredoxin family (Prxs) [97,98,99,100,101,102]. Extracellular superoxide dismutase (EC-SOD or SOD3) is a secretory, tetrameric 135 kDa glycoprotein consisting of two dimers similar to human SOD1 dimers. Superoxide dismutase is one of the most important enzymes protecting cells from the action of the superoxide radical anion, which converts radicals into molecular oxygen and hydrogen peroxide [103,104,105]. Superoxide dismutase has several isoforms containing various metal ions in its catalytic center and differing from each other in localization, the active center’s structure, and the structural organization of molecules [106]. Cu,Zn-SOD (SOD1; 32 kDa), containing two subunits, being the main cytosolic isoform of SOD, is also found in the intermembrane space of mitochondria and the nucleus. Mitochondrial Mn-SOD (SOD2), containing a manganese atom in the active center, is a critical enzyme for cell function [107,108]. Most of the Mn-SODs found in bacteria or other prokaryotes are dimers, and those found in the chloroplast matrix of eukaryotes, as well as in the mitochondria of mammals, including humans, are tetramers [108]. Each EC-SOD subunit contains one zinc and one copper and catalyzes the same reaction as the enzyme encoded by SOD1 [109]. EC-SOD is present in extracellular fluids, in the interstitial spaces of tissues, as well as milk, plasma, synovial fluid, and lymph. In addition, it is found in the nucleus of cells associated with chromatin. Thus, even though EC-SOD is mainly located in the extracellular space, this enzyme also has an intracellular localization [110].

Catalase is an enzyme capable of efficiently catalyzing of hydrogen peroxide decomposition into water and molecular oxygen. One catalase molecule can convert about 6 million H_2_O_2_ molecules per minute [111]. Catalase is primarily an intracellular enzyme. In cells, it is localized mainly in peroxisomes [112]. The highest catalase concentration is observed in liver erythrocytes and liver and kidney cells [113].

The family of glutathione peroxidases (GPxx) contains a large number of isoenzymes that catalyze the reduction of various organic hydroperoxides (ROOH) and hydrogen peroxide (H_2_O_2_). Currently, eight isozymes of mammalian glutathione peroxidases (GPx1, GPx2, GPx3, GPx4, GPx5, GPx6, GPx7, and GPx8), encoded by the GPX1-8 genes, have been identified. Despite the fact that the presence of this enzyme in the body is ubiquitous, the level of expression of each isoform varies depending on the type of tissue. Classical cellular glutathione peroxidase (GPx1 or cGPx), found in the cytosol and mitochondria, is a homotetramer that interacts with H_2_O_2_ and soluble low molecular weight hydroperoxides [114,115]. Each enzyme subunit contains one selenium atom, which is part of the selenocysteine residues in the active center [116]. GPx1 catalyzes the peroxide reduction reaction in the presence of glutathione (GSH). In brain mitochondria, in which GSS (glutathione synthetase) is not expressed, GPx1 can also use γ-glutamylcysteine (a precursor of GSH) as a cofactor [117]. GPx2 is mainly expressed in cells of the gastrointestinal tract, epithelial cells of the esophagus, and the liver [118]. The highest content of this enzyme is found in the bases of the villi of the large and small intestine [119]. GPx3 is the principal extracellular isoenzyme of the GPx family. It is found in various fluids such as milk, colloidal thyroid fluid, amniotic fluid, and bronchoalveolar lavage fluid. It was also found that GPx3 is contained in the basement membranes of cells of the proximal tubules of the kidney, membranes of intestinal epithelial cells, and lungs [120]. GPx4 exists in three different isoforms: cytosolic (cGPx4), mitochondrial (mGPx4), and nuclear GPx of sperm (snGPx4). This enzyme is present in most tissues and is contained both in the cytosol and membrane fraction. Depending on the availability of glutathione, GPx4 can act as both glutathione peroxidase and thiol oxidase [121]. GPx5 is a cysteine-containing glutathione peroxidase specific for the appendages of mice, rats, pigs, monkeys, and humans. The protein is found in epithelial cells and the lumen of the testes [122]. GPx6 is a selenocysteine-containing enzyme in humans and a cysteine-containing enzyme in rodents [123]. Data on this glutathione peroxidase are very limited. GPx7 is very similar in structure to GPx4; it contains in the active site cysteine instead of selenocysteine in the active site [123]. GPx8, another GPx family member, was found during phylogenetic analysis in mammals and amphibians [124]. This enzyme is a membrane protein of the endoplasmic reticulum.

Glutathione peroxidases, such as superoxide dismutase, are “adaptive” enzymes. Of the eight GPx isozymes, five (GPx1, GPx2, GPx3, GPx4, and GPx6) contain a selenocysteine residue, and three (GPx5, GPx7, and GPx8) use cysteine [125,126] in the active site. Their activities can sharply increase under conditions of oxidative stress; as a result, the foci of intense lipid peroxidation in the cell are limited and eliminated.

Peroxidases are enzymes widespread in biological systems. The family of mammalian peroxidases includes the following enzymes: myeloperoxidase (MPO), expressed in neutrophils [127]; eosinophilic peroxidase (EPO) found in eosinophils [128]; thyroid peroxidase (TPO), mainly localized in the thyroid gland [129]; and lactoperoxidase (LPO), found in milk, saliva, tears, and other exocrine fluids [130]. It was shown that LPO, MPO, EPO, and TPO are 46–57% identical in the amino acid residues sequence [131]. The primary function of them is the oxidation of various organic and inorganic electron-donor substrates in the presence of hydrogen peroxide [132,133].

Despite the fact that these enzymes are not directly related to antioxidant defense enzymes, the involvement of peroxidases in many processes with hydrogen peroxide assigns them an important role in regulating the ROS level.

Thus, the overproduction of reactive oxygen species is prevented by a complex antioxidant defense system consisting of antioxidants of both enzymatic and non-enzymatic nature.

## 5. Natural Antibodies with Oxidoreductase Activities

The first polyclonal immunoglobulins with catalytic activities were described in the work of Kulberg and Petyaev [134]. The authors demonstrated that IgG catalyzes superoxide-dependent processes in model chemical systems. In addition, it was shown that the activity of antibodies increases significantly upon the formation of an immune complex, possibly due to a change in the conformation of the antibodies molecules. Later, the catalytic properties of IgG in complex with R-proteins—products of catabolism of cell membrane receptors—were studied. As it turned out, both the complex itself and its individual components IgG and R—proteins had an enzymatic activity similar to that of superoxide dismutase. Still, the complex activity was 100 times higher [135].

The first catalytically active antibodies with proteolytic activity were found in the blood of patients with bronchial asthma [136,137]. They effectively bound the vasoactive intestinal peptide (*K*_m_ value ranged from 2.2 × 10^−6^ to 3.8 × 10^−8^ M) and had a high specificity. The authors have proved that the light chains of Abs contain the active centers of abzymes. The heavy chain of immunoglobulin is responsible for increasing the affinity upon binding of the substrate [137].

Many catalytic antibodies (IgGs and/or IgAs, IgMs) hydrolyzing different RNAs, DNAs, nucleotides, lipids, oligopeptides, proteins, and oligosaccharides were revealed in the sera of patients with various autoimmune and viral diseases (for review see [138,139,140,141,142,143,144,145,146,147,148,149,150,151,152,153] and refs therein). Then IgGs splitting DNAs were found in blood sera of patients with systemic lupus erythematosus (SLE) [138]. The third natural abzymes were IgGs of SLE patients with RNase activity [139]. Over the past 20–30 years, many antibodies with a wide variety of redox activities have also been discovered and described for the first time in this review.

### 5.1. Abzymes with Superoxide Dismutase (SOD) Activity

In 2000, the Lerner group carried out a number of experiments showing that antibodies have superoxide dismutase activity: immunoglobulins interacted with highly reactive singlet ^1^O_2_ and reduced it to a more stable O_2_^•−^ with the formation of hydrogen peroxide [154]. They used monoclonal Abs of various mammals belonging to IgG1k, IgG2ak, IgG2bk isotypes, F(ab)_2_ goat monoclonal antibodies, and polyclonal IgGs of horse and human. In many experiments, UV irradiation and a chemical source of ^1^O_2_ were used. For example, incubation of sheep monoclonal antibodies in the dark at 37 °C in the presence of 3′3′-(1,4-naphthylidene)-dipropionate also led to the formation of H_2_O_2_. As a possible reaction mechanism, the authors proposed the oxidation of aromatic amino acid residues or disulfide groups of the antibody molecule during the transfer of electrons to ^1^O_2_ [154]. In subsequent studies, the authors found that during the reduction of singlet oxygen, hydrogen peroxide is formed in higher concentrations than could be expected during the oxidation of the amino acids of the active center. Studies on the incorporation of isotopes and analysis of the kinetic parameters of the reaction made it possible to establish that antibodies use H_2_O as a source of electrons in singlet oxygen reduction [155]. As a result of this process, H_2_O_3_ is formed as the first intermediate product of the cascade of chemical reactions, which ultimately leads to the formation of H_2_O_2_. The primary function of the abzyme as a catalyst can be to stabilize the intermediate product H_2_O_3_ and prevent its reverse transformation. Datta D. and co-authors, based on special experiments, proposed a possible mechanism of ^1^O_2_ reduction [156,157].

From a detailed study of the Fab fragment active site, the interaction sites of the intermediate reaction products with the antibody were determined [156]. Three binding sites (S) for the H_2_O_3_ monomer and dimer (S1, S2, and S3) were found. S1 and S2 were located at the junction of the VH and VL chains, and one, S3, was between the CH1 and CL domains. The H_2_O_2_ binding site consisted of two clusters, P1 and P2. P1 is located at the base of the antigen-binding region; P2 is between the CL and CH1 domains and overlapped with S3. When studying the crystal structure of these clusters, no water molecules were found, which indicates that these sites were immersed in hydrophobic pockets. Trp109 was located near the S1 site of the heavy chain, which, according to the authors, may be responsible for the interaction with singlet oxygen.

The superoxide dismutase activity of polyclonal IgGs of Wistar rats was determined using an indirect method to inhibit the reduction of cytochrome C in the xanthine/xanthine oxidase system [158]. IgGs of all rats demonstrated a reliably tested activity. SOD activity of polyclonal IgGs from the serum of healthy donors and patients with acute reactive arthritis associated with urogenital infection caused by *Chlamydia trachomatis* was compared [159]. A statistically significant increase in the level of abzyme activity was found in patients compared with healthy individuals. The specific activity of Abs increased with an increase in the severity of the disease. The authors proposed that increased superoxide dismutase activity in various pathologies may be a protective anti-inflammatory mechanism.

A study of the SOD activity of IgGs was carried out in healthy people and patients with various types of multiple sclerosis (MS) [160]. An important feature of this work was the confirmation that the SOD activity is an intrinsic property of antibodies using a number of strict criteria. It was shown that SOD in patients with MS statistically significantly exceeds this activity in healthy people by 2–4 times, and the maximum activity was found in patients with relapsing–remitting MS. Based on the inhibitory analysis, the authors assumed the participation of copper ions in the catalysis of the superoxide anion radical dismutation by IgGs.

### 5.2. Catalytic Immunoglobulins with Catalase-like (CAT) Activity

In the past few years, the catalase activity of polyclonal immunoglobulins in patients with various diseases has been actively studied. In 2017, all the necessary experiments for the first time were performed to prove the assignment of catalase activity directly to antibodies, and showed, using IgGs of healthy donors and patients with schizophrenia, that human polyclonal IgGs possess catalase activity [161]. IgGs of patients with schizophrenia (36.4%) and from healthy donors (33.3%) possess catalase activity. Although the relative IgG catalase activity of schizophrenic patients and healthy donors varied significantly from patient to patient, it showed that the IgG activity of healthy donors is on average 15.8 times lower than that of schizophrenic patients. As indicated above, the activity of classical catalases depends on iron ions [162,163,164,165]. In the course of an enzymatic reaction leading to the destruction of H_2_O_2_, the iron ion bound to protoporphyrin IX [163] in the enzyme is first oxidized and then reduced by a second molecule of hydrogen peroxide according to following equations.
Catalase (Porf-Fe^III^) + H_2_O_2_ → Complex I (Porf• + -FeIV = O) + H_2_O(7)
Complex I (Porf• + -FeIV = O) + H_2_O_2_ → Catalase (Porf-FeIII) + H_2_O + O_2_(8)
Total reaction: H_2_O_2_ + H_2_O_2_ → 2H_2_O + O_2_

The authors investigated the metal and pH dependences and the kinetic parameters of the hydrogen peroxide decomposition reaction catalyzed by electrophoretically homogeneous IgGs. It showed that the obtained IgG preparations do not contain impurities of any classical enzymes, including canonical catalases. After extensive dialysis of IgGs against EDTA, the relative catalase activity of IgG preparations decreases on average ~2.5–3.7-fold; all IgGs demonstrated both independent and metal-dependent catalase activity. At the same time, a specific ratio of relative activity in the presence of ions of various metals was observed for each individual preparation of IgG. For example, Figure 1 demonstrates the data on the relative activity of three preparations in the presence of various metal ions [161].

In contrast to canonical catalases, Co^2+^ ions increased the reaction rate better than any other metal ions, while the effect of Cu^2+^, Mn^2+^, Ni^2+^, and Fe^2+^ ions on the activity of abzymes was weak or absent altogether. Each of the polyclonal IgG preparations exhibited several distinct pH optima ranging from 4.0 to 9.5. Figure 2 shows the pH dependence of catalase activity for four individual preparations of schizophrenic patients [161]. These data indicate that polyclonal antibody IgG preparations contain many monoclonal antibodies with very different properties and that repertoires of abzymes with catalase activity are unique in each person’s case. Catalase activity of human IgGs could probably, together with canonical catalases, play an important role in protecting mammalian organisms from oxidative stress and toxic compounds.

Catalase activity of IgGs and IgAs of patients with acute reactive arthritis [166], early arthritis [167], and patients with breast neoplasms [168] has been described. Interesting results were obtained from the study of IgG catalase activity in patients with various types of multiple sclerosis [160]. It showed that catalase activity of IgGs from healthy donors, on average, was two times lower than in patients with relapsing–remitting MS and patients with secondary progressive MS. At the same time, the authors did not find a statistically significant difference between the levels of IgG catalase activity in patients with different types of multiple sclerosis.

### 5.3. Catalytic Abs with Peroxidase and Peroxide-Independent Oxydoreductase Activities

For the first time, the analysis of H_2_O_2_-dependent peroxidase and H_2_O_2_-independent oxidoreductase activities of polyclonal antibodies with complete proof of the belonging of these activities to immunoglobulins was carried out using IgGs from blood sera of healthy Wistar rats [169,170]. It was shown that IgGs of Wistar rats oxidize 3,3′-diaminobenzidine (DAB), one of the classical substrates of horseradish peroxidase [169,170,171]. A distinctive feature of the study was the evidence that both observed IgG activities are the intrinsic function of antibodies and not a consequence of contamination of IgG preparations with any microcontaminants of enzymes. The strict criteria used may be summarized as follows [171]: (1) electrophoretic homogeneity of Abs; (2) FPLC gel filtration of antibodies using conditions dissociating strong noncovalent protein complexes in an acidic buffer (pH 2.6, Figure 3A) did not lead to a disappearance of peroxidase and oxidoreductase activities, and the peaks of two activities track exactly with IgGs; (3) immobilized rabbit antibodies against rat IgGs completely absorbed both peroxidase and oxidoreductase activities, and these activities were found only in the peak of Abs eluted by an acidic buffer (Figure 3B). The IgGs were subjected to SDS-PAGE to exclude possible artifacts due to hypothetical traces of canonical enzymes. Their activities were detected after incubating the gel in the reaction mixture containing DAB (Figure 3). The colored bands of the oxidation product of DAB were revealed only in positions corresponding to IgGs (for example, Figure 3C).

The *k*_cat_ values of the DAB oxidation by polyclonal IgGs isolated from the blood of different rats varied in the range from 1.8 × 10^2^ to 2.9 × 10^3^ min^−1^ in the presence of H_2_O_2_, and from 1.1 × 10^3^ to 3.6 × 10^3^ min^−1^ in the absence of H_2_O_2_. The *K*_m_ value in the presence and absence of H_2_O_2_ varied within (1.0–6.0) × 10^−4^ M. Thus, it was shown that the peroxidase and oxidoreductase activities of polyclonal IgG in Wistar rats are 1–3 orders of magnitude higher than the specific activity of most of other known artificial and natural catalytically active antibodies.

As indicated above, the iron ion is involved in the catalysis of reactions catalyzed by different peroxidases. It has been shown that IgG abzymes of Wistar rats with peroxidase and oxidoreductase activities catalyze the oxidation of DAB only in the presence of different metal ions [172]. Using the atomic emission method with excitation of spectra in a two-jet arc plasmatron, several divalent metal ions (Fe ≥ Pb ≥ Zn ≥ Cu ≥ Al ≥ Ca ≥ Ni ≥ Mn > Co ≥ Mg) were found bound with homogeneous IgG preparations obtained by standard methods [169,170,171]. All IgG preparations completely lost their activity after removing metal ions associated with them using EDTA [172]. Several external metal ions activated significantly both peroxidase and oxidoreductase activities of non-dialyzed against EDTA IgGs containing different internal metals showing pronounced biphasic dependencies corresponding to ~0.1–2.0 and ~2–5 mM of Me^2+^. For example, Figure 4 demonstrates several typical dependencies.

Cu^2+^ alone significantly stimulated dialyzed Abs’s peroxidase and oxidoreductase activities only at high concentrations (≥2.0 mM) (Figure 4A,B). In comparison, Mn^2+^ ions weakly activated peroxidase activity at concentration >3.0 mM but were active in the oxidoreductase oxidation at a low concentration (<1 mM) (Figure 4C,D). Fe^2+^-dependent peroxidase activity of dialyzed IgGs was observed at 0.1–5.0 mM, but it was completely inactive in the oxidoreductase reaction (Figure 4E,F). Ca^2+^, Mg^2+^, Zn^2+^, Al^2+^, Co^2+^, and Ni^2+^ were not capable of activating dialyzed Abs but slightly activated non-dialyzed IgGs. The use of the combinations of Cu^2+^ + Mn^2+^, Cu^2+^ + Zn^2+^, Fe^2+^ + Mn^2+^, and Fe^2+^ + Zn^2+^ led to a conversion of the biphasic curves observed for non-dialyzed IgGs to hyperbolic ones and in parallel to a significant increase in the activity as compared with Cu^2+^, Fe^2+^, or Mn^2+^ ions taken separately; the rates of the oxidation reactions, catalyzing by dialyzed non-dialyzed and IgGs, began to be comparable (Figure 5).

Mg^2+^, Ni^2+^, and Co^2+^ moderately activated the Cu^2+^-dependent oxidation of DAB catalyzed by dialyzed IgGs when Ca^2+^ inhibited these reactions. Interestingly, despite the presence of general regularities in the dependence of the rates of DAB oxidation, the ratio of these parameters for polyclonal IgGs from individual rats was noticeably different (Figure 4 and Figure 5). This indicated that the blood of each rat contains a different set of monoclonal IgGs that exhibit maximum activity in the presence of different metal ions.

It is known that some enzymes for their functioning use two metal ions. For example, Cu^2+^ ion participates in substrate oxidation in Cu, Zn-dependent superoxide dismutase. In contrast, the Zn^2+^ ion is a unique cofactor, which is necessary to stabilize the structure of the enzyme amino acid residues involved in the process of catalysis [173]. Thus, each of the second metal ions can serve as a second electrophilic metal cofactor of IgG catalytic centers supplementing the main variable oxidation state similar to the Zn^2+^ cofactor of Cu, Zn-dependent superoxide dismutase. At the same time, the observed strong increase in DAB oxidation by dialyzed IgG preparations in the presence of only one of the metal ions, and only at increased concentrations (Figure 4), may be because the second molecule of the same metal ions can act as a cofactor, but only at higher concentrations. Thus, in contrast to canonical peroxidases depending on iron ions, the pairs Cu^2+^ + Mn^2+^, Cu^2+^ + Zn^2+^, Fe^2+^ + Mn^2+^, and Fe^2+^ + Zn^2+^ turned out to be the most optimal for catalyzing the oxidation of the 3,3’-diaminobenzidine substrate by rat IgG antibodies. A small fraction was obtained by affinity chromatography of polyclonal IgGs on the Chelex 100 charged with metal ions. It showed that the specific activity of this fraction of rat antibodies in DAB oxidation is comparable to that for horseradish peroxidase [171].

The exceptional variety of abzymes with redox functions in the blood of rats is also evidenced by the data on the optimal pH values at which catalysis by polyclonal antibodies occurs. Figure 6 demonstrates as examples pH dependencies of peroxidase activity for IgGs of four rats [174]. One can see that all of the dependencies are remarkably different.

In contrast to classical plant and mammalian peroxidases, rat IgGs are more universal in their metal dependence and substrate specificity. Horseradish peroxidase has a broad specificity concerning various substrates [175]. It was shown that rat IgGs with peroxidase and oxidoreductase activities could efficiently oxidize DAB, phenol, *o*-phenylenediamine, α-naphthol p-hydroquinone, and NADH, but, in contrast to horseradish peroxidase, cannot oxidize adrenalin [176]. In comparison to IgGs, horseradish peroxidase cannot oxidize phenol, *p*-hydroquinone, and α-naphthol in the absence of H_2_O_2_.

The first results on the presence of antibodies with peroxidase activity in the blood of people were obtained by Generalov’s group [177]. IgG1, IgG2, and IgG4 isolated from the blood of patients with acute viral hepatitis B, chronic viral hepatitis C, and healthy donors in the presence of H_2_O_2_ oxidize aromatic amine. It was shown that the addition of CuSO_4_ at a final concentration of 47 µM led to a significant increase in activity compared to the initial one (on average 7.8 ± 3.0 times) [178]. The authors did not reveal a statistically significant difference in the activities of antibodies of healthy donors and patients with hepatitis. At present, it is evident that reliable results were obtained, but evidence of these activities belonging to antibodies was not presented in [177,178].

Later, IgG peroxidase activity was revealed in patients with chronic gastritis and duodenitis with the persistence of *Helicobacter pylori* in the antrum [179] and patients with gastric cancer and bronchial asthma [180]. It was found that in patients with autoimmune pathology of the thyroid gland, the peroxidase activity of IgGs is significantly higher than in healthy donors [180]. The authors suggested that in such patients, it may arise due to an anti-idiotypic mechanism, where the thyroid endoantigen, thyroperoxidase, acts as an inducer of abzimogenesis [180]. It also showed that the peroxidase activity of polyclonal IgAs from the oral fluid of patients with chronic periodontitis is significantly higher than the IgA activity of persons without periodontal pathology [181]. However, there was no proof of the belonging of peroxidase activity directly to antibodies in all these studies [177,178,179,180,181].

In 2015, the first detailed study of the enzymatic properties and substrate specificity of IgG from healthy donors with peroxidase and peroxide-independent oxidoreductase activities was carried out [182]. The results obtained indicate that these activities of antibodies are an intrinsic property of polyclonal IgGs isolated from the blood serum of healthy people. Using a set of strict criteria [17,146,149], described above in the case of antibodies from rat sera (for example, Figure 3), excluded the possibility of artifacts due to contamination of canonical enzymes. One of the most important criteria is detecting abzyme activities in in situ analysis. If this criterion is met, then all other criteria are also met. Figure 7 demonstrates that after SDS-PAGE of IgGs and incubation of the gel in a reaction mixture containing DAB, forming a colored oxidation product occurs only in the area corresponding to IgGs, and there are no other protein bands and areas of formation of a brown product.

In addition, it showed that only Fab- and F (ab)_2_—fragments of IgGs catalyze the oxidation of DAB [182]. As indicated above, abzymes from the blood of Wistar rats are absolutely dependent on metal ions in the oxidation of substrates and ultimately lose their activity after their removal. A completely different situation was found for abzymes from the blood of healthy people [182]. For example, Figure 8 shows data on a decrease in the activity of one IgG preparation of humans after removing metal ions, which are associated with antibodies isolated by the standard method.

After removing metal ions bound to antibodies, their peroxidase peroxide-independent oxidoreductase activity decreased, depending on the analysis of seven IgG preparations, from 100 to 10–85% and 100 to 14–83%, respectively, but did not completely disappear. IgG preparations after removal of metal ions were analyzed for metal composition by two-jet arc plasmatron atomic emission. According to the data obtained, they did not contain metals. The conclusion was drawn that human IgGs contain metal-dependent and metal-independent subfractions of abzymes with peroxidase and oxidoreductase activities [182].

It has been shown that the best IgGs activators of healthy donors in oxidation reactions are Cu^2+^ and Mn^2+^ ions [182]. In contrast to rat antibodies [172], other metal ions weakly or very weakly activated antibodies from human blood. The ratio of metal-dependent and metal-independent peroxidase and oxidoreductase activities of human IgGs turned out to be individual for each IgG preparation. It is known that glutathione peroxidase is a selenium-dependent enzyme; its activity is not associated with metal ions [114,183]. In addition, artificial monoclonal selenium-containing catalytic antibodies were obtained by replacing serine residues with selenocysteine in the active site [184,185,186]. These selenium-containing abzymes had glutathione peroxidase activity, mimicking glutathione peroxidase’s cytosolic form. The question arose as to the reason for human IgGs’ ability to catalyze the oxidation of substrates in the absence of metal ions, such as selenium-dependent glutathione peroxidase. Therefore, it can be assumed that metal-independent subfractions of human IgGs can be selenium-containing abzymes—analogs of glutathione peroxidase. By the method of X-ray fluorescence analysis using synchrotron radiation, it was shown that human IgG preparations contain 0.4–0.1 μg of selenium/1 g of IgGs [182]. Therefore, it can be assumed that metal-independent subfractions of human IgGs can be considered as selenium-containing abzymes—analogs of glutathione peroxidase.

As mentioned above, polyclonal IgGs from Wistar rats have broad substrate specificity [174,176]. Human IgGs are also capable of oxidizing various compounds. In the presence and the absence of H_2_O_2_, antibodies oxidized five substrates: 3,3′-diaminobenzidine (DAB), 2,2′-azino-bis (3-ethylbenzothiazoline-6-sulfonic acid) (ABTS), o-phenylenediamine (OPD), homovanilic acid (HVA), and *p*-hydroquinone (pHQ) [187]. Moreover, antibodies from the blood of patients with SLE and MS more efficiently oxidized various classical substrates of peroxidases [188].

Table 1 demonstrates values of catalytic constants (*k*_cat_, min^−1^) characterizing oxidation of these substrates with IgGs of healthy donors and patients with MS and SLE. It can be seen that the ranges of variation and the average values of *k*_cat_ for 10 IgG preparations of each of the three groups differ remarkably or significantly. In the case of most substrates, abzymes of SLE patients are most active. Only HVA was oxidized by Abs with peroxidase activity of healthy donors faster than by SLE IgGs (1.3-fold) and MS abzymes (2.4-fold). Three substrates: α-naphthol, 5-aminosalicylic acid (5-ASA), and 3-amino-9-ethyl carbazole (AEC), were effectively oxidized only by IgGs peroxidase activity of healthy donors and SLE patients (Table 2). Moreover, not all IgGs of these two groups had reliably tested peroxidase activity (Table 2). The average peroxidase activity of SLE IgGs statistically significant increase in comparison with Abs from healthy humans in the order (-times): OPD (1.2) < DAB (1.7) < α-naphthol (2.2) ≤ AEC (2.4) < ABTS (4.5) < 5-ASA (10.6), while with oxidoreductase activity: OPD (1.8) ≤ DAB (2.1-fold) < ABTS (5.0). The data show a tendency to the IgGs peroxidase and oxidoreductase activities increase in MS patients compared with healthy donors, but to a lesser extent: OPD (1.1–1.2-fold) ≤ ABTS (1.2–1.8-fold). It was previously shown that IgGs from Wistar rats are inactive in the oxidation of adrenaline [176]. Human IgGs also did not oxidize this substrate, both in the presence and absence of hydrogen peroxide. The average peroxidase activity of IgGs was 1.3, 1.5, and 2.6 times higher than the activity in the absence of H_2_O_2_ for substrates such as DAB, ABTS, and OPD, respectively [187].

These data indicate that the catalytic activity of abzymes with oxidative activities in humans increases with the development of autoimmune pathologies. However, in the development of AIDs, several stages are distinguished: onset, acute, and remission stages. In addition, the development of AIDs can be spontaneous or accelerated, induced by various antigens.

One of the models of human multiple sclerosis is C57BL/6 mice prone to spontaneous development of experimental autoimmune encephalomyelitis (EAE) [189,190]. It previously showed that the spontaneous development of EAE in C57BL/6 mice is associated with a specific change in the differentiation profile of hematopoietic stem cells in the bone marrow, cell apoptosis, and an increase in lymphocyte proliferation [191,192,193]. These slow changes during the spontaneous development of EAE are associated with the production of abzymes that hydrolyze DNA, myelin basic protein, MOG, and histones, the activity of which increases relatively smoothly as the pathology deepens. Immunization of mice with three different antigens (MOG_35–55_, DNA–histones complex, and complex of DNA with methylated bovine serum albumin (BSA)) led to accelerated development of EAE and an increase in the DNase and all proteolytic activities of abzymes [191,192,193]. After immunization of mice with antigens, three stages of EAE development are observed: onset (7–8 days), acute phase (18–20 days), and remission (>30 days). The maximum activity of abzymes in the hydrolysis of DNA, MBP, MOG, and histones depending on immunogen is observed in the acute phase of EAE development or later. Further study is required to understand whether antibodies, such as human IgGs, have redox functions and at what stage of EAE development they appear.

It was shown that IgGs of 3-month-old C57BL/6 mice, similar to DNase and protease activities, possess well detectable peroxidase and oxidoreductase activities. It has been shown that the best substrate for mouse antibodies is ABTS (2,2′-azino-bis(3-ethylbenzothiazoline-6-sulfonic acid) [194]. Therefore, ABTS was used to analyze overtime changes in two activities of IgGs during spontaneous EAE development [194]. Figure 9A shows that with the spontaneous development of EAE, the peroxidase activity of IgGs increases relatively slowly, and by 40 days, it increases approximately 1.7 times.

After immunization of mice with MOG, there is a sharp increase in the activity of abzymes in the hydrolysis of MOG (5.5-fold) and especially DNA (24.5-fold) in the acute phase (20 days) with a subsequent decrease in these activities (Figure 9B). At the same time, there is no sharp increase in peroxidase activity in the acute phase; it increases relatively smoothly: 5.3 times up to 40 days.

After immunization of mice with a complex of DNA–BSA, there is a slight increase in the activity of antibodies in the hydrolysis of MOG and DNA, approximately up to 20 days (Figure 9C). There is a delay in producing abzymes against DNA and MOG with a sharp increase after 20 days. To 40 days after immunization, DNase activity increases 122 times, and MOG-hydrolyzing activity—4.9 times (Figure 9C). Interestingly, the pattern of changes in the peroxidase activity of antibodies, in contrast to the spontaneous development and immunization by MOG, has a biphasic character similar to those for DNase and MBP-hydrolyzing activities, with an increase in peroxidase activity by day 40 and 3.5 times (Figure 9C).

Immunization of mice with a DNA–histone complex leads to a sharp parallel increase in DNase (14.4-fold), MOG-hydrolyzing (8.8-fold), and peroxidase (2.0-fold) activities by 20 days after immunization. Then these activities change over time in different ways (Figure 9D). Thus, the spontaneous development of EAE and its accelerated development after immunization of mice leads to an increase in all three activities, including peroxidase. However, each immunogen has a different effect on the pattern of the level of increase in the peroxidase activity of mice antibodies. The level of increase in the peroxidase activity of IgGs by 40 days decreases in the following order: 5.3-fold (immunization with MOG) > 3.5-fold (immunization with DNA–met–BSA) > 2-fold (immunization with DNA–histones) > 1.7-fold (spontaneous development EAE) (Figure 9).

## 6. Discussion

The reactive oxygen and nitrogen species play an important role in different diseases due to oxidative stress [195]. Oxidative disorders are a common part of many pathological processes, including autoimmune pathologies [195,196,197,198,199,200,201,202,203]. ROS initiates lesions by inducing blood–brain barrier disruption, reinforcing myelin phagocytosis, and leukocyte migration. They contribute to lesion persistence by mediating cellular damage and biological macromolecules important for functioning CNS cells [195,196,197,198,199,200,201,202,203].

Endogenous non-enzymatic and enzyme antioxidants usually countercheck oxidative stress. It showed that in MS, there is overexpression of superoxide dismutases, catalase, and heme oxygenase 1 [203]. Evidently, different antioxidant enzymes in all organisms are directed against oxidative stress. At the same time, antioxidant enzymes are mostly presented in different cells. Their catalytic activities in blood and various biological liquids are usually low since they quickly denature and hydrolyze, losing their activity [25,204]. Antibodies are stable in blood and remain intact for several months [205]. In addition, the concentration of antibodies in the blood is very high, including Abs with various oxidoreductase activities compared to canonical enzymes [140,141,142,143,144,145,146,147,148,149,150,151,152,153]. Thus, despite the fact that the specific activity of some abzymes with oxidoreductase activities can be 10–100 times lower than that of classical enzymes with the same activities, they can make a significant contribution to the protection of humans and animals from oxidative stress [169,170,171,172,182,187,188,194]. In addition, the blood and various biological fluids activity of many canonical oxidoreductases depends on the presence of heme and specific metal ions, most often iron ions [206]. Abzymes with such activities are independent of such cofactors and can function using a large number of ions of different metals.

As shown above, the appearance of abzymes with a wide variety of activities is a specific feature of autoimmune and some viral diseases with severe immune system disorders, including HIV infection [141,142,143,144,145,146,147,148,149,150,151,152,153].

In the blood of healthy donors, with rare exceptions, there are no abzymes that hydrolyze DNAs, peptides, and proteins. Exceptions are abzymes that hydrolyze vasoactive neuropeptide [137], thyroglobulin [207], and polysaccharides [208,209]. However, abzymes with such activities are found in the blood of a small percentage of conventionally healthy donors, and their activity is 10–1000 times lower than in patients with typical autoimmune diseases [140,141,142,143,144,145,146,147,148,149,150,151,152,153]. Unlike other abzymes, antibodies with redox functions have been found in the blood of all humans and animals examined without exception [154,155,156,158,159,160,161,166,167,168,169,170,171,172,176,177,178,179,180,181,182,187,188,194]. Interestingly, the highest activity of abzymes with peroxidase and peroxide-independent oxidoreductase activities was found in the blood of Wistar rats. Some fractions of rat antibodies, isolated by affinity chromatography on a sorbent charged with copper ions, exhibited specific activity comparable to horseradish peroxidase. Antibodies with peroxidase and peroxide-independent oxidoreductase activities from the blood of healthy donors were about 10 times less active abzymes from rats [187].

Wild rats are one of the most resistant animals to high doses of radiation and harmful chemicals. It is possible that in the process of adaptation to harsh, dangerous conditions of their existence, a transformation of their immune system occurred, which led to the formation of antibodies with increased activity in redox reactions. Interestingly, rat antibodies are extremely dependent on several different metal ions [172]. At the same time, human antibodies can catalyze redox reactions, both in the presence and absence of ions of metals [182].

Similar to artificial abzymes against stable analogs of transition states of catalytic reactions, naturally occurring abzymes are Abs raised against different enzymes substrates acting as haptens mimicking transition states of catalytic reactions [140,141,142,143,144,145,146,147,148,149,150,151,152,153]. While autoantibodies are produced by the B-cells of the immune system against various large molecules of DNAs, RNAs, proteins, etc., it is not entirely clear what immunogens are that stimulate the production of abzymes with redox functions. On the one hand, it could not be ruled out that the output of oxidoreductase abzymes may be stimulated by different mutagenic, carcinogenic, and other toxic compounds permeating into organisms of healthy humans. However, anti-idiotypic antibodies against active centers of various enzymes can also have catalytic activity [140,141,142,143,144,145,146,147,148,149,150,151,152,153]. It was shown that DNase Abzs of patients with AIDs present a “cocktail” of antibodies to DNA and RNA and anti-idiotypic antibodies against active centers of RNase, DNase I, DNase II, and other enzymes splitting nucleic acids [140,141,142,143,144,145,146,147,148,149,150,151,152,153]. In addition, abzymes with different redox activities in healthy humans and patients with AIDs can simply reflect the constitutive synthesis of germline Abs described by the Paul group [210,211].

For most classical components of living organisms, for example, proteins, there is a relatively simple phenomenon: one gene–one protein. However, according to theoretical estimates, against one antigen can be produced potentially up to a million antibodies with different properties. Wherein, the immune response to one and the same small and even large antigen may depend on what proteins or other components of blood it is associated with antigen [140,141,142,143,144,145,146,147,148,149,150,151,152,153]. All canonical enzymes usually catalyze only one reaction. In the case of Abs, the situation is entirely different. Against some antigens, antibodies can be produced with and without catalytic activity; of these, about 30–40% may be abzymes [141,153]. In addition, some anti-protein abzymes, such as against basic myelin proteins in MS and SLE patients, may be serine, thiol, or acid-type or metalloproteases [212,213,214]. However, there are abzymes in the active centers of which two active centers are combined, for example, serine and metalloproteases [215]. There are also antibodies, in which three activities are combined in the active centers—serine, metalloprotease, and DNase [216]. Moreover, the structure of these monoclonal antibodies and amino acid residues corresponding to two or three activities is very close to those for canonical enzymes with the same catalytic activities [215,216].

During the V (D) J recombination process, a unique DNA region encoding a variable domain is formed. The variable regions of heavy (H) or light (L) chains are encoded by a locus divided into several fragments—subgenes, which are designated V (from English variable), D (from English diversity), and J (from English joining) [217]. The V, D, and J subgenes encode the heavy chain variable region, while the light chain variable region encodes the V and J subgenres. After activation by antigens, B cells proliferate rapidly. With frequent loci divisions encoding the hypervariable domains of the light and heavy chains, in parallel, the enhancement in the frequency of point mutations, called somatic hypermutation, is observed. Somatic hypermutation usually occurs with a frequency of about one mutated variable domain nucleotide per cell division [217]. Thus, the daughter cells during the division will produce Abs containing changes in variable domains. Somatic hypermutations, therefore, serve as an additional mechanism for an increase in the diversity of immunoglobulins and affects the affinity of Abs to the antigen [218,219]. Thus, the principles of DNA formation, encoding abzymes, allow the formation of a large number of antibodies and abzymes (up to a million) with a wide variety of properties, including Abzs catalyzing redox reactions.

Analysis of the substrate specificity of antibodies in oxidation reactions of various substrates was carried out using polyclonal IgGs from the blood of healthy rats [176] and humans [187], as well as patients with SLE and MS [188]. Interestingly, these antibodies oxidize a large number of substrates of classical peroxidases and peroxide-independent oxidoreductases [174]. On the one hand, it could not be ruled out that some monoclonal antibodies included in the pool of polyclonal Abs, such as canonical enzymes, are capable of oxidizing several different substrates. On the other hand, some monoclonal antibodies can oxidize one set of substrates, while others can oxidize the other. However, the combination of different monoclonal antibodies allows the oxidation, by total abzymes, of a large number of substrates [176,187,188]. It is not yet well understood why the activity of antibodies in the oxidation of substrates in patients with autoimmune diseases is statistically significantly higher, and the substrate specificity of abzymes is expanded compared to in healthy donors [187,188].

As mentioned above, abzymes with redox activities in the blood of SLE and MS patients are statistically significantly higher than in healthy donors [188]. However, to analyze these activities, IgG preparations were taken from the patient’s blood at different stages of the development of these pathologies. Taking this into account, it was interesting to understand how the peroxidase and oxidoreductase activities of antibodies can be changed during different stages of AIDs development. For this purpose, the analysis of in time possible changes in the peroxidase activity of C57BL/6 mice antibodies during the spontaneous and antigen-induced development of EAE was used.

It showed that spontaneous development in C57BL/6 mice of EAE is coupled with a specific change in bone marrow HSCs’ differentiation profile and lymphocyte proliferation in several organs [191,192,193]. Treatment of these mice with MOG, DNA–histones, and DNA–met–BSA leads to parallel acceleration of EAE and appearance of harmful Abzs hydrolyzing DNA, MOG, and MBP [191,192,193]. However, in parallel with the production of these abzymes, there is a gradual increase in the antibodies oxidizing standard substrates. A strong parallel increase in all enzymatic activities of IgGs, including the redox one, may speak in favor of the fact that with the spontaneous development of AIDs, an extended chance in the differentiation profile of HSCs can occur. In other words, such a multiple expansion of the stem cell differentiation profile stimulates the appearance of B-lymphocytes against at least several internal antigens. Immunization of mice with MOG and complexes of DNA with histones and met-BSA changes not only the production of specific abzymes that hydrolyze DNA, MBP, MBP, and histones, but also abzymes that oxidize substrates due to peroxidase and peroxide-independent oxidoreductase activities [194]. This also indicates that the activation of the production of B-lymphocytes, synthesizing abzymes with different activities, is somehow connected. A complete specific differentiation of stem cells and the production of abzymes occur already at the level of the cerebrospinal fluid of the bone marrow. Abzymes with different catalytic activities from the cerebrospinal fluid of patients with multiple sclerosis are approximately 30–60 times more active than from the blood of the same patients [220,221,222].

Considering the data discussed above, the question arises why antibodies with redox functions are needed in healthy people. It should emphasize again that the blood sera of healthy people and animals usually do not contain abzymes, except in catalyzing redox reactions. The emergence of many various abzymes is clearly associated with the development of various AIDs [145,146,147,148,149,150]. All data on abzymes with redox catalytic functions suggest that these antibodies can oxidize different compounds harmful to humans. Therefore, they can reduce oxidative disorders’ pathological effects and protect the body from oxidative stress. Due to the high content and prolonged circulation of IgGs with redox activities in blood they can decrease oxidative disorders in the bloodstream. In addition, the ability of Abs to be accumulated in the foci of inflammation, abzymes with redox activities, along with canonical catalases and glutathione peroxidases, can participate in the regulation of H_2_O_2_ concentration and limit the damage caused by a high concentration of hydrogen peroxide and oxidative processes in these zones.

Taking into account a set of many different compounds that are dangerous for mammals as substrates for IgGs with oxidoreductase activities [176,187,188], it is easy to propose that a set of possible substrates of catalytic antibodies can be extensive, including some drugs and xenobiotics. Some abzymes with superoxide dismutase activity can convert O_2_^−^ into hydrogen peroxide, while other Abs with peroxidase and catalase activities neutralize H_2_O_2_. Taken together, polyclonal Abzs can serve as an additional system of human and various animals of reactive oxygen species detoxification.

## 7. Conclusions

This review is the first to summarize the literature data on abzyme antibodies that catalyze redox reactions. For comparison, data on the features of the functioning of classical enzymes with the same redox functions are presented. It has been shown that abzymes and canonical enzymes in the blood of humans and animals are very different in a large number of parameters. In addition, the concentration of antibodies and the lifetime of antibodies in biological fluids are greater than that of the classic oxidoreductases, which are localized mainly in cells. Abzymes possess broad substrate specificity and oxidize many different compounds similar to canonical oxidoreductases. Thus, polyclonal abzymes can serve as an additional system of human and various animals reactive oxygen species detoxification and be important antioxidants.

## Figures and Tables

**Figure 1 ijms-23-03898-f001:**
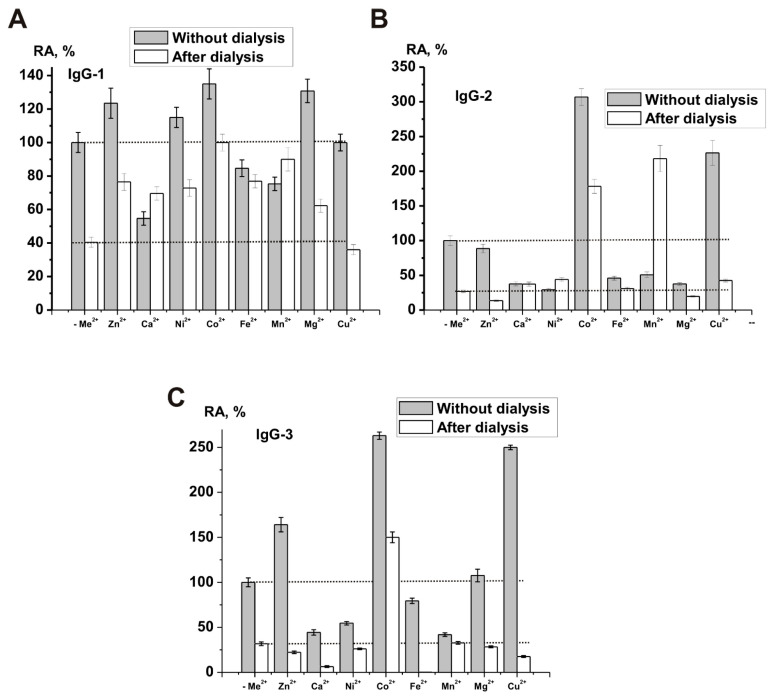
Effect of the dialysis of three individual IgG preparations from sera of schizophrenic patients against EDTA and different external metal ions on the relative activity (RA, %) of dialyzed and non-dialyzed preparations (**A**–**C**). The relative activity of every non-dialyzed preparation was taken for 100%. All IgGs and metal ions used are marked on Panels A–D [161].

**Figure 2 ijms-23-03898-f002:**
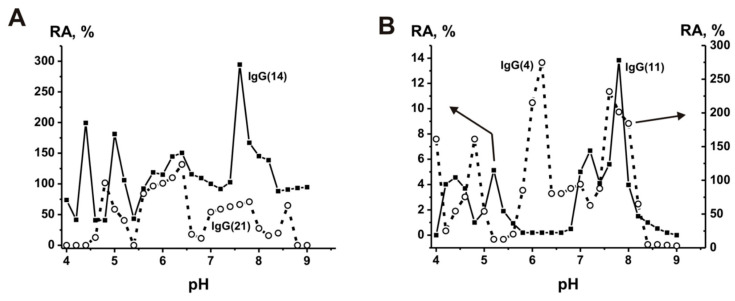
Dependences of the relative catalase activity of four schizophrenic patients’ IgGs on pH of the reaction mixture (**A**,**B**). All relative activities of IgG(14) at pH 7.0 were taken for 100 percent. The errors in the initial rate determination from three experiments in each case did not exceed 7–10% [161].

**Figure 3 ijms-23-03898-f003:**
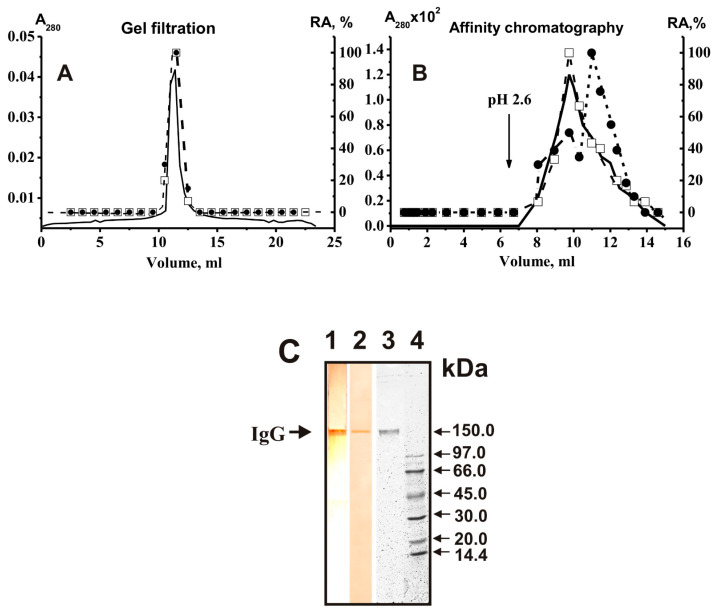
Gel filtration of rat IgGs on a Superdex 200 column in an acidic buffer (pH 2.6) after Ab incubation with this buffer (**A**) and affinity chromatography of IgGs on Sepharose bearing immobilized rabbit IgGs against rat IgGs (**B**): (—), absorbance at 280 nm; the relative activity of IgGs (RA, %) was measured using DAB in the presence (•) and the absence (□) of H_2_O_2_. In situ SDS-PAGE analysis of rat IgGs peroxidase and oxidoreductase activities (**C**). The peroxidase activity was revealed by incubation of longitudinal gel slices in the reaction mixture containing DAB, 10 mM H_2_O_2_, and 5 mM CuCl_2_ (lane 1, **C**), while peroxide-independent oxidoreductase activity—incubation without H_2_O_2_ (lane 2, **C**). An additional control longitudinal band of the same gel was stained with Coomassie R250 to reveal IgG positions (lane 3, **C**). Arrows show the positions of proteins’ molecular mass markers (lane 4, **C**) [171].

**Figure 4 ijms-23-03898-f004:**
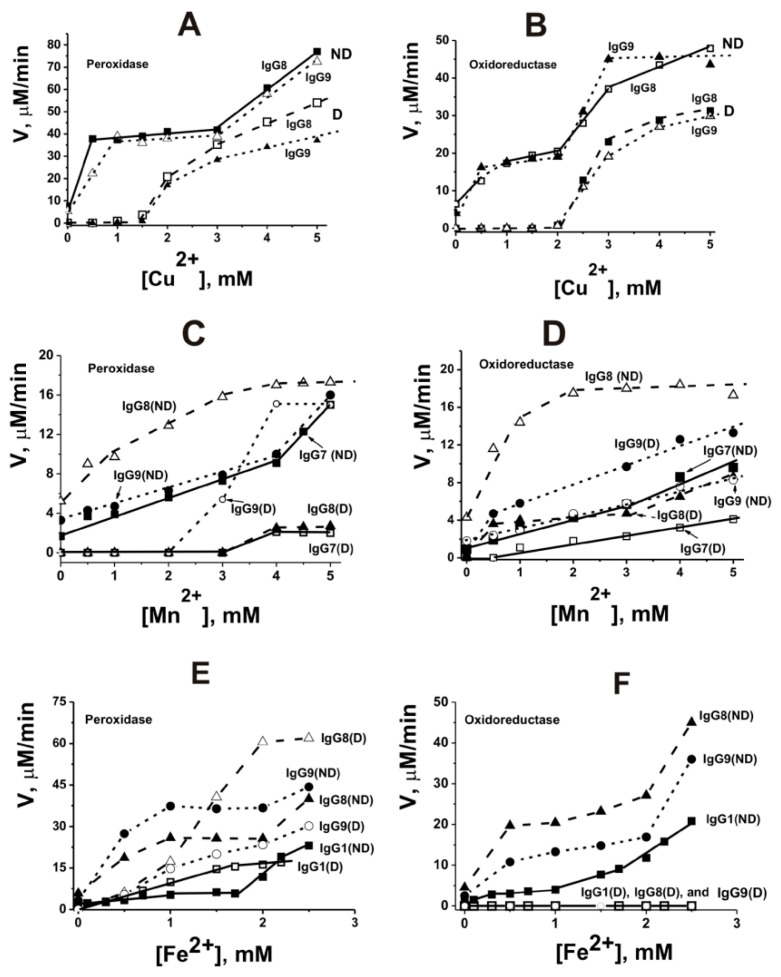
Dependencies of the peroxidase and oxidoreductase activities in the IgG-dependent oxidation of DAB (0.2 mg/mL) on the concentration of CuCl_2_ (**A**,**B**), MnCl_2_ (**C**,**D**), and FeCl_2_ (**E**,**F**). Curves corresponding to dialyzed (D) and non-dialyzed (ND) IgGs (5 µg/mL) and numbers of IgG preparations are marked in Panels **A**–**F** [172].

**Figure 5 ijms-23-03898-f005:**
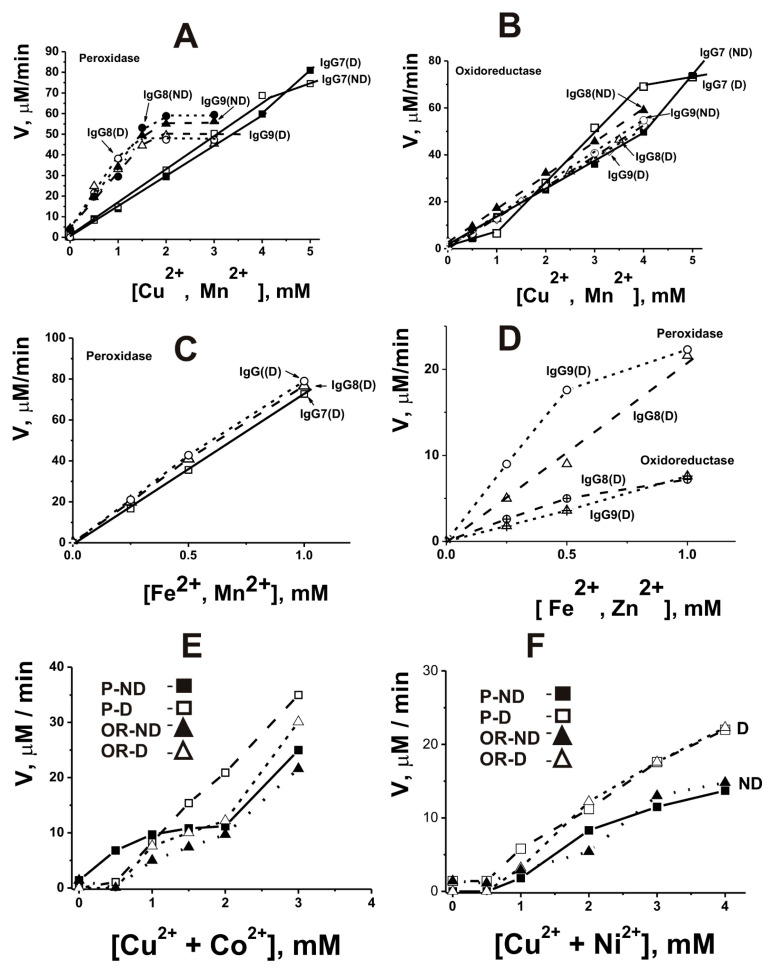
Dependencies of the peroxidase and oxidoreductase activities in the IgG-dependent oxidation of DAB (0.2 mg/mL) on the concentration of equimolar mixtures of MnCl_2_ with CuCl_2_ (**A**,**B**), FeCl_2_ with MnCl_2_ or ZnCl_2_ (**C**,**D**); CuCl_2_ with CoCl_2_ (**E**); NiCl_2_ (**E**); or NiCl_2_ (**F**). Curves corresponding to dialyzed (D) and non-dialyzed (ND) IgGs (5 µg/mL) and the numbers of IgG preparations are marked in Panels (**A**–**F**). Additional designations in (**E**,**F**) panels refer to non-dialyzed and dialyzed (P-ND and P-D) peroxidase and non-dialyzed and dialyzed (OR-ND and OR-D) oxidoreductase activities of IgGs [172].

**Figure 6 ijms-23-03898-f006:**
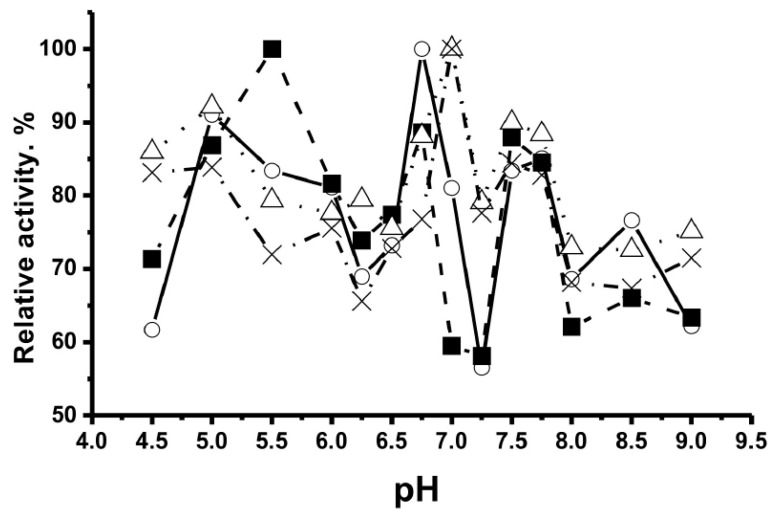
Dependences of the relative peroxidase activity of four Wistar rats IgGs (∆, □, ×, and o) on pH of reaction mixture The error in the initial rate determinations from three experiments in each case did not exceed 7–10% [174].

**Figure 7 ijms-23-03898-f007:**
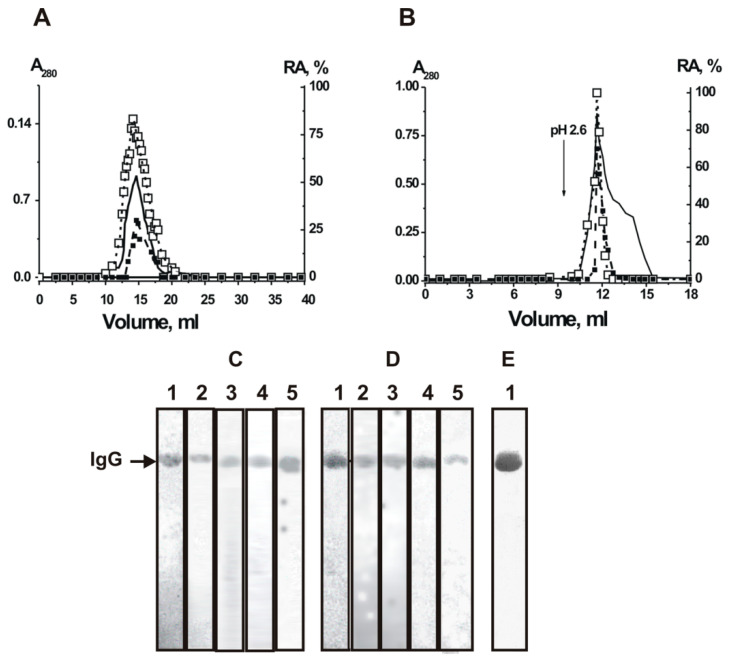
FPLC gel filtration of IgG_mix_ (mixture of 10 preparations) on a Superdex 200 column under acidic conditions (pH 2.6) after IgG_mix_ incubation in the same buffer (**A**) and affinity chromatography of IgG_mix_ on Sepharose bearing mouse IgGs against human IgGs (**B**): (—), absorption at 280 nm; relative activity (RA) of IgG_mix_ in peroxidase (□) and oxidoreductase oxidation (■) of DAB. Relative activity (RA) of the fractions with the highest activity was taken for 100%. In situ SDS-PAGE analysis of five different IgGs peroxidase (**C**, lanes 1–5; 10 µg of Abs) and oxidoreductase (**D**, lanes 1–5; 20 µg of Abs) activities from five different donors using a nonreducing 4–15% gradient gel (**C**,**D**) The oxidizing activity of IgGs was revealed by the appearance of a colored oxidation product after incubation longitudinal gel slices in the reaction mixture containing DAB and H_2_O_2_ (**C**) or without hydrogen peroxide (**D**). Longitudinal slices of the gels were stained with Coomassie R250 (**E**, lane 1) to reveal the positions of IgGs [182].

**Figure 8 ijms-23-03898-f008:**
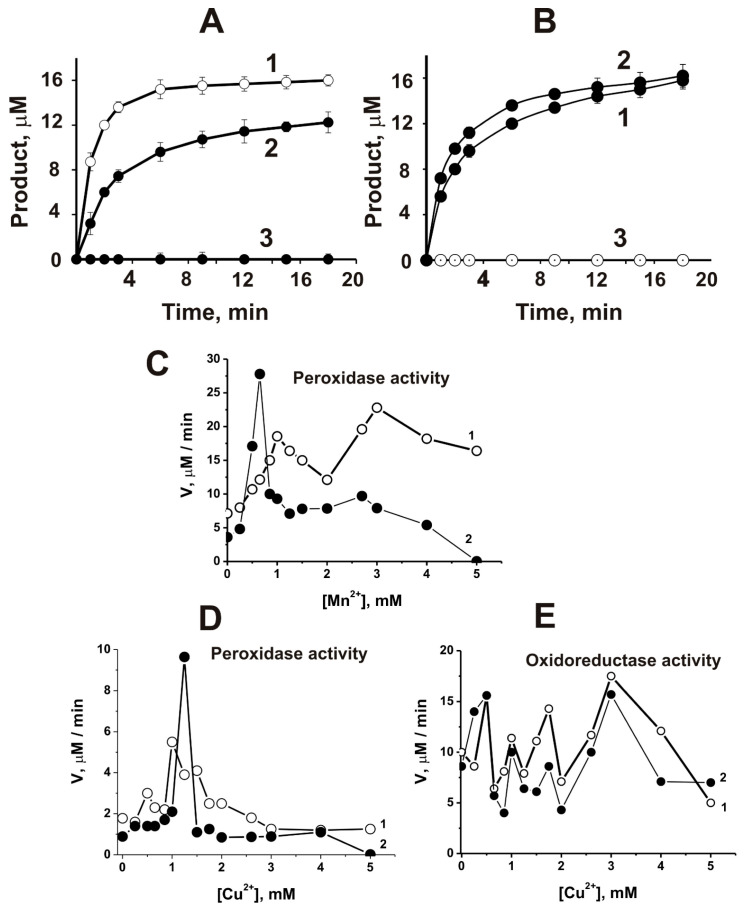
The time-dependences of DAB oxidation in the peroxidase (**A**) and oxidoreductase (**B**) oxidation of the substrate catalyzed by IgG3 before (curve 1) and after (curve 2) its extensive dialysis against 0.1 M EDTA + 0.1 M EGTA and then against 0.3 M EDTA; curve 3, control oxidation of DAB in the absence of Abs. The dependencies of relative peroxidase activity of IgG9 (28 nM) on concentrations of MnCl_2_ (**C**); peroxidase (**D**) and oxidoreductase (**E**) activities of IgG10 (20 and 120 nM, respectively) on the concentration of CuCl_2_. These IgG9 and IgG10 were used before (curve 1) and after (curve 2) their dialysis against EDTA + EGTA [182].

**Figure 9 ijms-23-03898-f009:**
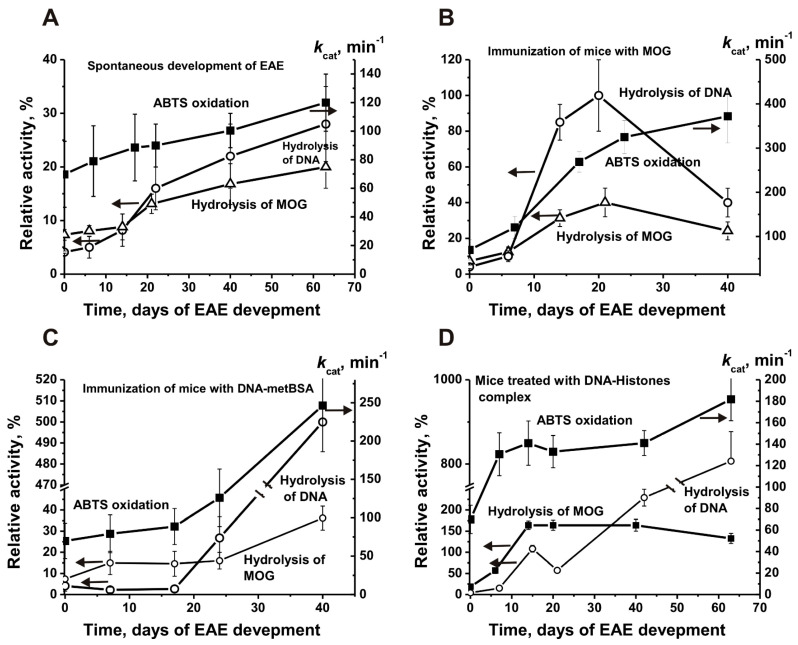
Over time changes in the relative activity of mice IgGs in the hydrolysis of MOG and DNA in the (**A**–**D**; left scales) as well as oxidation of ABTS (*k*_cat_, min^−1^; **A**–**D**; right scales). The mean values of the activities of IgGs from seven mice during spontaneous (**A**) as well as accelerated development of EAE after immunization of mice with MOG (**B**), DNA–met–BSA (**C**), and DNA–histones (**D**) are given. All designations are marked in the panels: the arrows indicate to which *Y*-axis the given curve belongs—to the right or the left [194].

**Table 1 ijms-23-03898-t001:** The apparent *k_cat_* values characterizing peroxidase and oxydoreductase activity of individual IgGs from the sera of healthy donors, patients with SLE and MS in the oxidation of five substrates [188] *.

IgG Number	*k*_cat_, min^−1^ **					
	3,3’-diaminobenzidine (DAB)					
	Healthy Donors		SLE Patients		MS Patients	
	+H_2_O_2_	−H_2_O_2_	+H_2_O_2_	−H_2_O_2_	+H_2_O_2_	−H_2_O_2_
Range	51.6–92.8	41.8–58.0	56.3–154.0	134.4–165.1	30.1–70.7	21.8–61.8
Average value	66.3 ± 12.0	52.1 ± 5.3	114.0 ± 33.2	111.8 ± 31.0	45.5 ± 12.2	27.7 ± 12.0
2,2′-azino-bis(3-ethylbenzothiazoline-6-sulfonic acid) diammonium salt (ABTS)						
Range	54.7–97.4	36.6–69.5	170.1–655.2	170.2–430.8	95.0–174.6	19.4–98.2
Average value	73.7 ± 12.0	50.5 ± 12.9	355.0 ± 164.5	271.4 ± 89.4	134.3 ± 32.5	59.8 ± 26.7
o-phenylenediamine (OPD)						
Range	0.22–0.79	0.15–0.55	0.42–1.1	0.26–0.69	0.43–1.1	0.02–0.47
Average value	0.64 ± 0.20	0.25 ± 012	0.77 ± 0.17	0.46 ± 0.13	0.67 ± 0.19	0.3 ± 0.15
homovanillic acid (HVA)						
Range	11.5–27.9	0.0–0.0	0.13–1.1	0.0–0.0	0.24–0.31	0.0–0.0
Average value	19.8 ± 5.6	~0.0	0.50 ± 0.27	~0.0	0.28 ± 0.04	~0.0
*p*-hydroquinone (pHQ)						
Range	0.0–1.12	0.0–2.3	1.4–15.2	3.7–16.8	8.2–12.9	3.7–13.0
Average value	0.21 ± 0.42	0.36 ± 0.79	8.7 ± 4.9	9.3 ± 4.5	10.4 ± 2.4	7.4 ± 4.9

* For each value, a mean of three measurements was used; the error of the determination of values did not exceed 7–10%; the range and average values for 10 IgG preparations for each of the three groups are given. ** The apparent *k_cat_* values of the reaction at fixed concentrations of H_2_O_2_ (10 mM) and five substrates: DAB (0.93 mM), ABTS (0.364 mM), OPD (0.185 mM), HVA (0.549 mM), and pHQ (0.124 mM) were calculated: *k_cat_* = V (M/min)/[IgG], M. The maximum concentrations of substrates were used at which they are soluble and do not form precipitation of their oxidation products.

**Table 2 ijms-23-03898-t002:** The apparent *k_cat_* values characterizing peroxidase activity of individual IgGs from the sera of healthy donors, patients with SLE in the oxidation of α-naphtol, 5-ASA, and AEC [188] *.

IgG Number	Several Substrates in the Presence of H_2_O_2_; *k*_cat_, min^−1^ **					
	Healthy Donors			SLE		
	α-Naphtol	5-ASA	AEC	α-Naphtol	5-ASA	AEC
Range	0.0–26.9	0.0–2.0	25.2–48.9	6.3–39.7	0.4–16.4	65.2–155.2
Average value	8.6 ± 8.5	0.48 ± 0.65	38.0 ± 8.5	18.9 ± 11.6	5.1 ± 4.6	91.3 ± 25.9

* For each value, a mean of three measurements is reported; the error of the determination of values did not exceed 7–10%; the range and average values for 10 IgG preparations for each of the two groups are given. ** The apparent *k_cat_* values of the reaction at fixed concentrations of H_2_O_2_ (10 mM), α-naphtol (0.07 mM), 5-aminosalicylic acid (5-ASA, 0.38 mM), and 3-amino-9-ethylcarbazole (AEC, 0.19 mM) were calculated: *k_cat_* = V (M/min)/[IgG], M. The maximum concentrations of substrates were used at which they are soluble and do not form precipitation of their oxidation products.

## Data Availability

All data is given in this article.

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
