# Peer review of "Essential Protective Role of Catalytically Active Antibodies (Abzymes) with Redox Antioxidant Functions in Animals and Humans"

_ijms, 2022, doi:10.3390/ijms23073898_

Round 1
Reviewer 1 Report
The review article presents new scientific problem related to unexpected properties of antibodies, i.e. their catalytic activity with special function as endogenous antioxidants. The enzymatic activity of abzymes, active antibodies, was studied extensively by Authors of the review and their co-workers, and presented before in numerous experimental articles in well recognized international journals. The main expectation of their work was to find the potential role of abzymes exhibiting peroxidase and oxidoreductase activity in the protection of patients with autoimmune pathologies against oxidative stress. Very common approach to the protection of living organisms against free radicals and reactive oxygen/nitrogen species is almost exclusively based on the action of enzymatic antioxidants and low molecular weight endo- and exogenous antioxidants. However in many studies this barrier was shown to be not efficient enough in case of chronic and terminal diseases, especially. As Authors mentioned in the review, the activity of enzymatic antioxidants decreases with the progress of diseases due to wear or not sufficient synthesis of enzymatic proteins, deficiency of some trace elements essential for them (Fe, Cu, Zn, Mn), or other factors. The hypothesis for the studies assumed that the antibodies with redox activity might be powerful antioxidants and take over this function instead of endogenous enzymatic antioxidants, in advanced stages of autoimmune diseases.
The review is based mainly on numerous experimental studies done by Authors and their co-workers, which results were published earlier in the recognized journals. Introduction presents the overview of oxidative stress, sources of reactive species, the main enzymatic and non-enzymatic antioxidants. Chapter No. 5 contains the overview of antibodies with redox activities, i.e. the abzymes capable to replace superoxide dismutase, catalase, and peroxidase activity in scavenging ROS. This part of the article contains 9 figures published before in Authors paper within years. According to the title of Table 1 and Table 2, data presented there were published in reference [181], but by Furtmueller and co-workers in 2006. The question is whether Authors received the permission to publish it. It seems also that it may be a mistake in numbering of references (?).
The review is very interesting and well organized, contains many data and wide spectrum of the abzymes antioxidative properties studied in in vitro experiments. There are some emerging questions on the abzymes activity in vivo and their actual role in living organisms while intensified oxidative stress.
In Discussion, the significant contribution of the abzymes to the protection of humans and animals against oxidative stress is being considered, due to stability and quite high concentrations of antibodies in the blood. Next factor, i.e. independence of specific metal ion concentration may was mentioned to play an important role. Definitely, more experiments, and clinical studies in advanced stages of diseases accompanied by oxidative stress are necessary before acceptance of the antioxidative function of abzymes.
From chemical and biochemical point of view, it may be questioned the bond, positions in the structure - different chains - of antibodies responsible for binding the metal ions. All studies ions are transient metal ions, and may form covalent or coordination bonds with the amino acid side chain functional groups. Copper, iron or manganese ions participate in enzymatic redox processes by changing their oxidation status, thus the same role play probable while bonding to antibodies and are responsible for their antioxidant activity.
In Conclusion, Authors summarize the broad substrate specificity of abzymes and point the differences in many parameters with antioxidant enzymes. The hypothesis of polyclonal abzymes function as antioxidants needs more studies, especially clinical, but seems to be very promising.
The review is supported by wide set of references, some of them are very recent and basic to the topic, and give sufficient overview of described issues.
Author Response
Referee 1
The review article presents new scientific problem related to unexpected properties of antibodies, i.e. their catalytic activity with special function as endogenous antioxidants. The enzymatic activity of abzymes, active antibodies, was studied extensively by Authors of the review and their co-workers, and presented before in numerous experimental articles in well recognized international journals. The main expectation of their work was to find the potential role of abzymes exhibiting peroxidase and oxidoreductase activity in the protection of patients with autoimmune pathologies against oxidative stress. Very common approach to the protection of living organisms against free radicals and reactive oxygen/nitrogen species is almost exclusively based on the action of enzymatic antioxidants and low molecular weight endo- and exogenous antioxidants. However in many studies this barrier was shown to be not efficient enough in case of chronic and terminal diseases, especially. As Authors mentioned in the review, the activity of enzymatic antioxidants decreases with the progress of diseases due to wear or not sufficient synthesis of enzymatic proteins, deficiency of some trace elements essential for them (Fe, Cu, Zn, Mn), or other factors. The hypothesis for the studies assumed that the antibodies with redox activity might be powerful antioxidants and take over this function instead of endogenous enzymatic antioxidants, in advanced stages of autoimmune diseases.
The review is based mainly on numerous experimental studies done by Authors and their co-workers, which results were published earlier in the recognized journals. Introduction presents the overview of oxidative stress, sources of reactive species, the main enzymatic and non-enzymatic antioxidants. Chapter No. 5 contains the overview of antibodies with redox activities, i.e. the abzymes capable to replace superoxide dismutase, catalase, and peroxidase activity in scavenging ROS. This part of the article contains 9 figures published before in Authors paper within years. According to the title of Table 1 and Table 2, data presented there were published in reference [181], but by Furtmueller and co-workers in 2006. The question is whether Authors received the permission to publish it. It seems also that it may be a mistake in numbering of references (?).
Answer: Sorry this is mistake in the designation of references. It was corrected for [188] according to corrected References . As for other tables and figures, they are all taken from open access journals. In the case of such journals, permission to use Figures and Tables in other articles and reviews is not required.
The review is very interesting and well organized, contains many data and wide spectrum of the abzymes antioxidative properties studied in in vitro experiments. There are some emerging questions on the abzymes activity in vivo and their actual role in living organisms while intensified oxidative stress.
In Discussion, the significant contribution of the abzymes to the protection of humans and animals against oxidative stress is being considered, due to stability and quite high concentrations of antibodies in the blood. Next factor, i.e. independence of specific metal ion concentration may was mentioned to play an important role. Definitely, more experiments, and clinical studies in advanced stages of diseases accompanied by oxidative stress are necessary before acceptance of the antioxidative function of abzymes.
Answer: In general, you are right, there are still not enough such experiments in the literature on the analysis of protection against oxidative stress by abzymes. Additional experiments and clinical studies in the advanced stages of diseases accompanied by oxidative stress would be very useful. However, this is a very difficult question for investigation. Only on the basis of clinical studies in the later stages will also not be enough for an absolutely reliable conclusion. It is necessary to somehow compare the initial and final stages of a large number of patients and see that those who have a higher concentration of abzymes antioxidants and less depth of the disease. However, in the case of each patient, other individual diseases may be mixed in. With this in mind, this kind of analysis with some possibility will be carried out in the future probably using different mice.
From chemical and biochemical point of view, it may be questioned the bond, positions in the structure - different chains - of antibodies responsible for binding the metal ions. All studies ions are transient metal ions, and may form covalent or coordination bonds with the amino acid side chain functional groups. Copper, iron or manganese ions participate in enzymatic redox processes by changing their oxidation status, thus the same role play probable while bonding to antibodies and are responsible for their antioxidant activity.
The binding of abzymes to metal ions was analyzed using the example of monoclonal antibodies with DNase and metal-dependent protease activities. Analysis of DNA sequences and corresponding protein sequences showed that metal-dependent abzymes contain specific protein sequences for metal ion chelation, the same as canonical metal-dependent DNases and proteases. It should be assumed that a similar situation takes place in the case of abzymes with redox functions. If oxidation with the participation of metal ions as a result of their binding occurred not in the specific active centers, but somewhere in other parts of the antibodies, their activity would be low and not comparable with that for canonical oxidoreductases, as is observed in the case of rat antibodies. In control experiments with other proteins such as albumin, which are capable of binding metal ions, it was shown that the activity is not significantly increased compared to free metal ions in solution.
In Conclusion, Authors summarize the broad substrate specificity of abzymes and point the differences in many parameters with antioxidant enzymes. The hypothesis of polyclonal abzymes function as antioxidants needs more studies, especially clinical, but seems to be very promising.
The review is supported by wide set of references, some of them are very recent and basic to the topic, and give sufficient overview of described issues.
Thaks a lot for comments
Sincerely
Prof. Georgy A. Nevinsky
Reviewer 2 Report
The Authors of this review article discuss the properties of catalytically active antibodies (abzymes) with antioxidant functions, in comparison with well-known enzymes involved in the detoxication of reactive oxygen and nitrogen species. This is a useful and informative review article. The Authors are invited, though, to improve some weaknesses of their paper with appropriate additions and/or corrections, particularly for the purpose of improving the general scientific background of the article.
Major comments:
Title:
please rephrase the title as follows:
“Essential protective role of catalytically active antibodies (abzymes) with redox antioxidant functions in animals and humans”. Please also correct “people” to “humans” wherever it recurs in the text body.
Terminology:
I consider the use of the adjective “natural”, with referral to catalytically active antibodies (abzymes) in the following expressions inappropriate and possibly misleading:
- “natural antibodies” in line 18 of the Abstract,
- “natural antibodies with oxidoreductase activities” in the Title of Section 5.
- “natural abzymes were IgGs of SLE patients with RNase activity” in Section 5, line 253
- “natural catalitically active antibodies” in Paragraph 5.3, line 394
- “naturally occurring abzymes” in line 704 of the Discussion
The designation “natural antibodies” is broadly accepted as indicating antibodies, mostly of the IgM isotype, directed towards a wide range of ubiquitous natural antigens, both of self and non-self origin. The majority of these circulating IgM are polyspecific autoantibodies, secreted in a T-cell-independent manner and encoded by multiple germ-line variable region genes, with little somatic mutation. They have been implicated in the maintenance of peripheral immune tolerance. It is my opinion that the Authors of the present review article should abstain from indicating the abzymes they are discussing as “natural antibodies”: this designation seems inappropriate both when referring to artificial antibodies, i.e., antibodies obtained as a result of immunization practices, and to antibodies found in the sera and other biological fluids of patients affected by autoimmune diseases, such as SLE.
Section 2 (“Reactive oxygen and nitrogen species (RONS) etc.”):
- lines 101-107:
the Authors should discuss more clearly and orderly, with appropriate references and reaction formulas when indicated, the following points:
1.1. the formation of anion superoxide radical (O2•-) in cells during aerobic metabolic processes and its conversion to H2O2 and O2 by superoxide dismutase
1.2. the conversion of part of the H2O2 thus formed, by reaction with the O2•- radical, to •OH in vivo through the iron-catalyzed Haber-Weiss reaction
1.3. the formation of •OH radicals, under physiological conditions, also by the Fe(II)- or Cu(II)-catalyzed cleavage of H2O2 in the Fenton reaction.
In regard, they may cite the following references:
Stadtman, E. R., Berlett, B.S. Reactive oxygen-mediated protein oxidation in aging and disease. Chem. Res. Toxicol. 10(5): 485-494, 1997.
Stadtman, E. R., Levine, R. L. Free radical-mediated oxidation of free amino acids and amino acid residues in proteins. Amino Acids 25: 207-218, 2003
1.4. the myeloperoxidase-catalyzed generation of hypochlorous acid (HOCl) from chloride (Cl−) and hydrogen peroxide (H2O2), HOCl being implicated in the pathogenesis of several chronic inflammatory cardiovascular and pulmonary diseases and cancer.
In regard, they may cite the following references:
Maitra D, Shaeib F, Abdulhamid I, et al. Myeloperoxidase acts as a source of free iron during steady-state catalysis by a feedback inhibitory pathway. Free Radic Biol Med. 2013;63:90-98.
1.5. the formation of •OH radical from the reaction of Fe(II) ions with HOCl
- The Authors please use consistently the •OH symbol
- lines 106-107:
the Authors should mention that both the hydroxyl radical (•OH) and the superoxide anion radical (O2•-) with its protonated form, the hydroperoxyl radical (HO2•), can be generated in aqueous solutions by the use of X-, g- or UV-rays and can attack electrophilically both the peptide main chain and the side chains of peptides, polypeptides and proteins, also causing proteolytic cleavage.
In regard, they may cite the following references:
Schuessler, H., Schilling, K. Oxygen effect in radiolysis of proteins. Part 2. Bovine serum albumin. Int. J. Radiat. Biol. 45: 267-281, 1984
Garrison, W. M. Reaction mechanisms in the radiolysis of peptides, polypeptides, and proteins. Chem. Rev. 87(2): 381-398, 1987
- lines 122 and following.
The Authors overlook and should describe instead, although briefly, the consequences of the formation of adducts of lipid peroxidation (LPO) products, such as the reactive aldehydes MDA and HNE, with the cell proteome. There is a vast body of evidence that LPO-mediated damage to multiple proteins with essential functions causes cell dysfunction and death in a broad range of degenerative, chronic inflammatory, autoimmune and neoplastic diseases in humans.
In regard, they may cite the following references:
Pizzimenti S, Ciamporcero E, Daga M, Pettazzoni P, Arcaro A, Cetrangolo G, Minelli R, Dianzani C, Lepore A, Gentile F, Barrera G. Interaction of aldehydes derived from lipid peroxidation and membrane proteins. Front Physiol. 2013 Sep 4;4:242
Barrera G, Pizzimenti S, Ciamporcero ES, Daga M, Ullio C, Arcaro A, Cetrangolo GP, Ferretti C, Dianzani C, Lepore A, Gentile F. Role of 4-hydroxynonenal-protein adducts in human diseases. Antioxid Redox Signal. 2015 Jun 20;22(18):1681-702
Section 3 (“Non-enzymatic antioxidants”)
This section should be fortified with more information concerning vitamin E (a-tocopherol), vitamin C (l-ascorbate), Vitamin A (retinol) and lipoic acid.
Author Response
Referee 2
Comments and Suggestions for Authors
The Authors of this review article discuss the properties of catalytically active antibodies (abzymes) with antioxidant functions, in comparison with well-known enzymes involved in the detoxication of reactive oxygen and nitrogen species. This is a useful and informative review article. The Authors are invited, though, to improve some weaknesses of their paper with appropriate additions and/or corrections, particularly for the purpose of improving the general scientific background of the article.
Major comments:
Title:
please rephrase the title as follows:
“Essential protective role of catalytically active antibodies (abzymes) with redox antioxidant functions in animals and humans”. Please also correct “people” to “humans” wherever it recurs in the text body.
Answer: It was corrected (all changes in the article in red)
Terminology:
I consider the use of the adjective “natural”, with referral to catalytically active antibodies (abzymes) in the following expressions inappropriate and possibly misleading:
Считаю использование прилагательного «натуральный» применительно к каталитически активным антителам (абзимам) в следующих выражениях неуместным и, возможно, вводящим в заблуждение:
- “natural antibodies” in line 18 of the Abstract,
- “natural antibodies with oxidoreductase activities” in the Title of Section 5.
- “natural abzymes were IgGs of SLE patients with RNase activity” in Section 5, line 253
- “natural catalitically active antibodies” in Paragraph 5.3, line 394
- “naturally occurring abzymes” in line 704 of the Discussion
The designation “natural antibodies” is broadly accepted as indicating antibodies, mostly of the IgM isotype, directed towards a wide range of ubiquitous natural antigens, both of self and non-self origin. The majority of these circulating IgM are polyspecific autoantibodies, secreted in a T-cell-independent manner and encoded by multiple germ-line variable region genes, with little somatic mutation. They have been implicated in the maintenance of peripheral immune tolerance. It is my opinion that the Authors of the present review article should abstain from indicating the abzymes they are discussing as “natural antibodies”: this designation seems inappropriate both when referring to artificial antibodies, i.e., antibodies obtained as a result of immunization practices, and to antibodies found in the sera and other biological fluids of patients affected by autoimmune diseases, such as SLE.
Answer: It was corrected (in red in text)
Section 2 (“Reactive oxygen and nitrogen species (RONS) etc.”):
- lines 101-107:
the Authors should discuss more clearly and orderly, with appropriate references and reaction formulas when indicated, the following points:
- the formation of anion superoxide radical (O2-) in cells during aerobic metabolic processes and its conversion to H2O2and O2 by superoxide dismutase
Answer: It was corrected (in red in text)
- the conversion of part of the H2O2thus formed, by reaction with the O2- radical, to •OH in vivo through the iron-catalyzed Haber-Weiss reaction
Answer: It was corrected (in red in text)
- the formation of OH radicals, under physiological conditions, also by the Fe(II)- or Cu(II)-catalyzed cleavage of H2O2in the Fenton reaction.
- Answer: It was corrected (in red in text)
1.4. the myeloperoxidase-catalyzed generation of hypochlorous acid (HOCl) from chloride (Cl−) and hydrogen peroxide (H2O2), HOCl being implicated in the pathogenesis of several chronic inflammatory cardiovascular and pulmonary diseases and cancer.
In regard, they may cite the following references:
Answer: It was corrected (in red in text)
1.5. the formation of •OH radical from the reaction of Fe(II) ions with HOCl
- The Authors please use consistently the •OH symbol
- lines 106-107:
the Authors should mention that both the hydroxyl radical (•OH) and the superoxide anion radical (O2•-) with its protonated form, the hydroperoxyl radical (HO2•), can be generated in aqueous solutions by the use of X-, g- or UV-rays and can attack electrophilically both the peptide main chain and the side chains of peptides, polypeptides and proteins, also causing proteolytic cleavage.
Answer: It was corrected (in red in text)
- lines 122 and following.
The Authors overlook and should describe instead, although briefly, the consequences of the formation of adducts of lipid peroxidation (LPO) products, such as the reactive aldehydes MDA and HNE, with the cell proteome. There is a vast body of evidence that LPO-mediated damage to multiple proteins with essential functions causes cell dysfunction and death in a broad range of degenerative, chronic inflammatory, autoimmune and neoplastic diseases in humans.
Answer: It was corrected (in red in text)
Section 3 (“Non-enzymatic antioxidants”)
This section should be fortified with more information concerning vitamin E (a-tocopherol), vitamin C (l-ascorbate), Vitamin A (retinol) and lipoic acid.
Answer: It was corrected (in red in text)
Thank you very much for very useful comments? We tried to add all infioration, which you ask
With best wishes
Sincerely
Prof. Georgy A. Nevinsky
Round 2
Reviewer 2 Report
The Authors addressed conveniently all the points of criticism that I had raised in my previous review of their article.